# Phase separation by the polyhomeotic sterile alpha motif compartmentalizes Polycomb Group proteins and enhances their activity

Elias Seif[1], Jin Joo Kang[1,2], Charles Sasseville[1], Olga Senkovich[3], Alexander Kaltashov[1], Elodie L. Boulier[1], Ibani Kapur[1,2], Chongwoo A. Kim[3] & Nicole J. Francis [1,2,4 ✉]

Polycomb Group (PcG) proteins organize chromatin at multiple scales to regulate gene expression. A conserved Sterile Alpha Motif (SAM) in the Polycomb Repressive Complex 1 (PRC1) subunit Polyhomeotic (Ph) has been shown to play an important role in chromatin compaction and large-scale chromatin organization. Ph SAM forms helical head to tail polymers, and SAM-SAM interactions between chromatin-bound Ph/PRC1 are believed to compact chromatin and mediate long-range interactions. To understand the underlying mechanism, here we analyze the effects of Ph SAM on chromatin in vitro. We find that incubation of chromatin or DNA with a truncated Ph protein containing the SAM results in formation of concentrated, phase-separated condensates. Ph SAM-dependent condensates can recruit PRC1 from extracts and enhance PRC1 ubiquitin ligase activity towards histone H2A. We show that overexpression of Ph with an intact SAM increases ubiquitylated H2A in cells. Thus, SAM-induced phase separation, in the context of Ph, can mediate large-scale compaction of chromatin into biochemical compartments that facilitate histone modification.

[1] Institut de recherches cliniques de Montréal, 110 Avenue des Pins Ouest, Montréal, QC H2W 1R7, Canada. [2] Division of Experimental Medicine, McGill University, 1001 Decarie Boulevard, Montreal, QC H4A 3J1, Canada. [3] Department of Biochemistry and Molecular Genetics, Midwestern University, 19555N. 59th St., Glendale, AZ 85308, USA. [4] Département de biochimie et médecine moléculaire Université de Montréal, 2900 Boulevard Edouard-Montpetit, Montréal, QC H3T 1J4, Canada. ✉email: nicole.francis@ircm.qc.ca

Polycomb group (PcG) proteins repress gene expression by modifying chromatin at multiple scales, ranging from post-translational modification of histone proteins to organization of megabase- scale chromatin domains[1–5]. Two main PcG complexes, PRC1 and Polycomb Repressive Complex 2 (PRC2), are central to PcG function and conserved across evolution[1–4]. Both complexes can carry out post-translational modification of histones (methylation of histone H3 on lysine 27 (H3K27me) for PRC2 and ubiquitylation of lysine 118/119 of histone H2A (H2A-Ub) for PRC1). PRC1, and to a lesser extent, PRC2, are also implicated directly in long-range organization of chromatin and clustering of PcG proteins into foci in cells[6–11]. Two classes of PRC1 complexes have been defined, canonical (cPRC1) and noncanonical (ncPRC1). Both types of complexes contain two ring-finger proteins required for E3 ubiquitin ligase activity toward H2A (Psc and dRING in *Drosophila*, Pcgf and Ring1A or B in mammals)[3,4]. cPRC1 additionally contains a Cbx protein (Pc in *Drosophila*), and a PHC (Ph in *Drosophila*). In ncPRC1, RYBP replaces the Cbx protein, PHCs are absent, and other accessory proteins are variably present, depending on the Pcgf subunit[4]. At least in mouse embryonic stem cells, ncPRC1 is responsible for the bulk of ubiquitylated H2A[12–14]. This suggests that histone modification and chromatin organization may be partitioned between nc and cPRC1s, although both types of complexes share many genomic targets[12,13,15,16]. All cPRC1 subunits can interact with DNA and/or chromatin, and both canonical and ncPRC1s can compact chromatin in vitro[15,17], but polyhomeotic (Ph), and thus cPRC1, is the most implicated in large-scale chromatin organization[3,10,11,18–21].

Ph is a core subunit of canonical PRC1, and its most notable feature is the presence of a conserved sterile alpha motif (SAM) in its C terminus that can assemble into head-to-tail helical polymers[22]. SAMs are present in many different types of proteins and in many cases can mediate protein polymerization[23]. The SAM of Ph is required for Ph function in *Drosophila*, and its full polymerization activity is important for gene repression[24,25]. PRC1 forms visible foci both in *Drosophila* and in mammalian cells[7,11], and, in *Drosophila* cells, a much larger number of diffraction-limited clusters[10]. Disrupting the Ph SAM impairs formation of PcG protein clusters and reduces long-range contacts among PcG-bound loci, suggesting that the two processes are related[10,11]. Despite the wealth of in vivo data supporting the critical function of Ph SAM in large-scale organization of PcG proteins and chromatin, and in gene regulation, the biochemical mechanisms by which Ph SAM links protein and chromatin organization are not known.

In recent years, an important role for liquid–liquid-phase separation (LLPS) in organizing macromolecules in cells has been defined[26–29]. This mechanism is increasingly accepted as being important in formation of protein–RNA membraneless organelles[29,30], and has more recently been implicated in chromatin compartmentalization and genome organization[31–35], transcription activation[36–38], DNA repair[39,40], and PcG protein organization[41–43]. LLPS by nuclear-/chromatin- associated proteins may concentrate proteins and RNAs, enhance or inhibit reactions, exclude other factors, and even physically move genomic regions[26,28,44]. Nucleated phase separation at superenhancers mediated by disordered regions in transcription factors and coactivators is believed to be important for driving cycles of active transcription[36,37]. Phase separation is also implicated in the formation of heterochromatin and its function as a distinct chromatin environment[33,35], although the precise role of LLPS is debated[45]. The mammalian PcG protein Cbx2 (part of certain cPRC1s) has also been shown to undergo LLPS in vitro with chromatin, and to form foci in mammalian cells, suggesting a link between LLPS and PcG function[41,43].

Here, we consider the hypothesis that Ph SAM can organize chromatin through phase separation by analyzing Ph-chromatin interactions in vitro. We find that a truncated version of Ph containing the SAM, HD1, and FCS domains connected by a disordered linker forms phase-separated condensates with chromatin or DNA. Condensate formation depends on Ph SAM, and is facilitated by its polymerization activity. Ph SAM-dependent condensates can recruit PRC1 components from extracts, and enhance the ubiquitin ligase activity of PRC1 toward histone H2A. In cells, overexpressed Ph forms foci, and increases H2A-Ub levels. Thus, phase separation is an activity of Ph SAM that can condense chromatin and enhance PRC1 activity.

## Results

**A truncated version of Ph, Mini-Ph, forms phase-separated condensates with DNA or chromatin.** In *Drosophila melanogaster*, the *Ph* gene is present as a tandem duplication in the genome; the two genes (*Ph-p* and *Ph-d*) encode highly related proteins with largely redundant function[46]. *Drosophila* Ph is a large protein (1589 amino acids for Ph-p), the majority of which is disordered (Fig. 1a), and which is difficult to work with in vitro. To focus on the function of the domains conserved in Ph orthologs, particularly the SAM, and to facilitate biochemical analysis, we used a truncated version of *Drosophila* Ph-p, termed Mini-Ph[5]. Mini-Ph (aa1289–1577) contains the three conserved domains—from amino- to carboxyterminus: the HD1, the FCS zinc finger that can bind nucleic acids[47], and the Ph SAM (Fig. 1a). An unstructured linker connects the FCS to the SAM, and restricts Ph SAM polymerization[5]. Thus, while Ph SAM alone forms extensive helical polymers in vitro, Mini-Ph exists mainly as short polymers of 4–6 units (Fig. 1b, c), even at high concentrations[5].

We expressed Mini-Ph in *Escherichia coli*, purified it (Supplementary Fig. 1A), and tested whether it can form phase-separated condensates, alone or with chromatin. Chromatin was prepared on a circular plasmid containing 40 copies of the *Lytechinus* 5S rDNA nucleosome-positioning sequence[48] using histone octamers fluorescently labeled with Cy3 on histone H2A (Supplementary Fig. 1C–E). Neither Mini-Ph alone, nor chromatin alone form condensates in buffer (Fig. 1d, e). When Mini-Ph is mixed with chromatin, or plasmid DNA, large, round, phase-bright drops are observed (Fig. 1f; Supplementary Fig. 2A). Drops formed with either DNA or chromatin undergo fusion (Fig. 1g, Supplementary Fig. 2B; Supplementary Movies 1–3), and settle to the bottom of the imaging plate where they flatten and continue to fuse (Fig. 1h). Under phase-separation conditions, Mini-Ph and DNA can be pelleted by centrifugation (Supplementary Fig. 2C, D), consistent with them forming a denser phase. Mini-Ph–DNA solutions also become turbid, as measured by $OD_{340}$ (Supplementary Fig. 2E). To evaluate the relationship between the concentration of chromatin or DNA and Mini-Ph, and phase separation, we titrated both Mini-Ph and DNA or chromatin, and manually scored each point in the resulting matrix as one or two phases (Fig. 1i, j; Supplementary Fig. 2F, G). This produces a limited coarse-grained delineation of the boundary between one- and two-phase regimes. Phase separation is sensitive to the concentration of both components, and the ratio between the two. This is most notable for Mini-Ph–DNA titrations, where we are able to add high concentrations of DNA, which prevent phase separation (Supplementary Fig. 2F, G). From similar titrations of NaCl and Mini-Ph at a fixed DNA concentration, we find that phase separation is observed in NaCl concentrations up to 125 mM (Supplementary Fig. 3). We conclude that Mini-Ph forms phase-separated condensates with either DNA or chromatin.

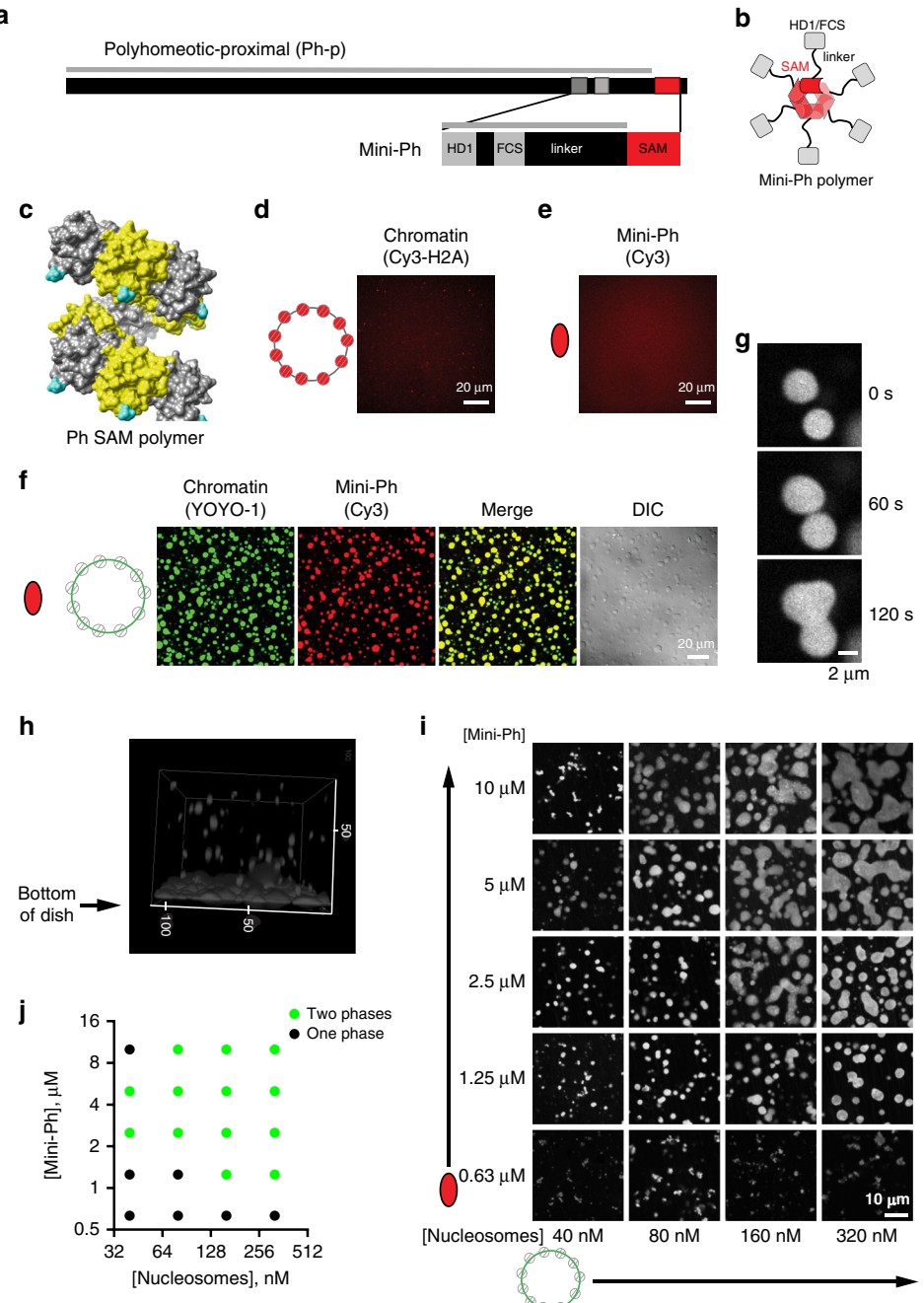

**Fig. 1 Mini-Ph forms phase-separated condensates with chromatin. a** Schematic of Polyhomeotic–proximal and Mini-Ph, which spans aa 1389–1577 and includes the 3 conserved domains and an unstructured linker. The gray line indicates predicted disordered sequence (using PONDR-VSL2)[92]. Note that 91.9% of the sequence is predicted to be disordered (disregarding segments less than 30 amino acids), with only the SAM predicted to be ordered. **b** Schematic depicting the oligomeric state of Mini-Ph, which forms limited polymers of 4–6 units (6 are shown)[5]. **c** Structure of nine units of the Ph SAM polymer demonstrating its helical architecture. The N terminus, from which the linker extends, is shown in cyan. PDB 1D 1KW4. **d**, **e** Neither chromatin (**d**) nor Mini-Ph (**e**) form condensates in buffer. **f** Mini-Ph forms phase-separated condensates with chromatin. This observation was repeated with three different preparations of Mini-Ph and more than ten different chromatin preparations. **g** Time lapse of droplet fusion of Mini-Ph-chromatin condensates, visualized with Alexa-647-labeled Mini-Ph. **h** 3D reconstruction of confocal stack of images demonstrating that Mini-Ph-chromatin condensates form a fused layer on the bottom of the imaging plate. Scale is in microns. **i** Representative images from a matrix of Mini-Ph and chromatin showing the relationship between protein and chromatin concentration and condensate formation. [Nucleosomes] assume 8 fmol of nucleosomes per 1 ng of DNA. Images are representative of two independent experiments. **j** Graph depicting the conditions where one phase and two phases were scored in two experiments like the one shown in (**i**). See also Supplementary Figs. 1–3 and Supplementary Movies 1–3.

The disordered linker connecting Ph SAM to the FCS domain was previously demonstrated to restrict Ph SAM polymerization, possibly due to its ability to contact Ph SAM *in trans*[5]. A scrambled linker has the same effect, implicating amino acid composition rather than organization[5]. The sequence properties of linkers that connect a structured domain play a central role in phase separation[28], by restricting or promoting interactions between structured domains, and by contributing weak interactions[49]. We therefore analyzed the sequence properties of the linker (Supplementary Fig. 4), both in *Drosophila* Ph, and in the three human homologs (PHC1–3). The Ph linker is acidic (pI 3.9), but relatively uncharged (fraction-charged residues (FCR) = 0.15), and does not have strongly segregated charge (Supplementary Fig. 4B, E, Supplementary Data 1). Overall, the Ph linker is expected to be collapsed (Supplementary Fig. 4D).

The linker region is conserved between the two *Drosophila* Ph homologs (Supplementary Fig. 4F), but both the sequence and charge properties of the linker in mammalian PHCs are distinct (Supplementary Fig. 4C–E, G; Supplementary Data 1). The human PHC linkers are basic (pI > 10), more charged (FCR: 0.25–0.34), have more segregated charges, and have a higher fraction of expansion-promoting residues (Supplementary Fig. 4C, E). They occupy a distinct position on the Das–Pappu diagram of states (Supplementary Fig. 4D), predicting context-dependent collapse or expansion. Previous analysis indicates that the PHC3 linker promotes polymerization of either PHC3 or Ph SAM, and does not interact with the PHC3 SAM *in trans*[5]. A synthetic linker designed to be unstructured (Rlink[5]) promotes polymerization of both Ph and PHC3 SAM, and shares properties with PHC linkers, including a basic pI (Supplementary Data 1). Evolutionary tuning of the linker sequences is likely to affect phase-separation properties of PHCs, although this will need to be tested experimentally.

**Chromatin is highly concentrated in Mini-Ph condensates**. One potential function of phase separation is to concentrate (compact) chromatin. To measure the concentration of chromatin in Mini-Ph-chromatin condensates, we first prepared calibration curves using the same Cy3-labeled histone octamers (labeled on H2A) that were used to assemble chromatin (Supplementary Fig. 5A). The concentration of nucleosomes in Mini-Ph condensates, starting from a mixture of 150 nM nucleosomes, and 5 μM Mini-Ph, was measured as 22.5 ± 4.4 μM (SD) (Supplementary Fig. 5B). We note that this value is lower than the reported concentration of chromatin in pure chromatin condensates induced by monovalent cations (~340 μM[31]). The reported measurements used free dye to prepare the calibration curve. When we imaged calibration curves prepared from free Cy3, although the curves are linear, they predict at least a 60× higher concentration than curves prepared with labeled histone octamers using the same imaging parameters. Because ladders prepared with free Cy3 do not accurately predict known concentrations of Cy3-labeled histone octamers in our hands, we believe that the chromatin concentrations measured using the Cy3-labeled histone-calibration curve (Supplementary Fig. 5) are correct for Mini-Ph–chromatin condensates.

**Mini-Ph is dynamic in condensates, but chromatin intermixes slowly**. A characteristic of liquid condensates is that the components are dynamic. We carried out fluorescent recovery after photobleaching (FRAP) experiments with Mini-Ph–chromatin condensates. A fraction of Mini-Ph is mobile and exchanges in condensates, so that bleached Mini-Ph drops partially recover fluorescence within several minutes (Fig. 2a, b; Supplementary Fig. 6A–D). In contrast, when the histones (labeled with H2A–Cy3)

were bleached, less than 15% of the fluorescence is recovered after several minutes (Fig. 2b, Supplementary Fig. 6E, F). We quantified our FRAP data with user-selected region of interest (ROI) for the bleach area and background, and fit the data with a double-exponential function (Eq. (1)). Recent work has drawn attention to the complexity of FRAP measurements in phase-separated condensates, and in selecting and applying the appropriate biophysical model to the data[50]. Because of the complexities cited above, we interpret the FRAP curves qualitatively. Although we have calculated the half-times of the fast and slow populations and mobile fractions from our data (Supplementary Fig. 6A–D), we do not think that these numbers can be used to compare with other systems, or with the measured FRAP behavior of Ph in vivo[51]. Nevertheless, they indicate that Mini-Ph and chromatin have very different kinetics in condensates. Similar behavior has been dissected in a model system of lysine or argnine-rich peptides and homopolymers of RNA[52]. In this case, slow kinetics for the RNAs could be explained by RNA–RNA interactions[52]. It is possible that nucleosome–nucleosome interactions contribute to the slow kinetics of chromatin. However, it must also be emphasized that the chromatin templates used in these experiments are large (11 kb of DNA, ~55 nucleosomes, ~13,750 kDa). This system may partially mimic chromatin in vivo, which also does not freely intermix (discussed in ref. [53]). Experiments with 12-nucleosome linear arrays (more than 4× smaller than the templates used here) indicate that while chromatin alone can form a liquid-like state that shows (slow) recovery in FRAP experiments[31], in most conditions, chromatin forms condensates that behave as solids and do not recover in FRAP experiments, similar to chromatin in vivo[54].

To further understand how chromatin intermixes in condensates, we used two-color chromatin experiments (Fig. 2c–h). Mini-Ph was incubated separately with chromatin labeled with Cy3 or Alexa 647. Once condensates had formed, the two sets were mixed together, and images collected as the condensates fused (Fig. 2c–h). Although condensates of both colors fused, ultimately forming a fused network at the bottom of the imaging plate (Fig. 2g, h), distinct Cy3 and Alexa-647 regions remained, indicating that the chromatin in preformed condensates does not fully intermix when the condensates fuse, at least over 60 min that we monitored (Fig. 2h). This is in clear contrast to control experiments in which the two chromatins are mixed prior to addition of Mini-Ph, where all structures contain a uniform mix of both fluorophores (Fig. 2c, d). These experiments are consistent with the coexistence of different dynamics in Mini-Ph-chromatin condensates. The persistence of unmixed regions could also reflect dynamically arrested phase separation in the preformed condensates. We note that in the mixtures shown in Fig. 2c–h, the Alexa-647-labeled chromatin (white in Fig. 2) has a slightly lower nucleosome density than the Cy3 (red)-labeled chromatin. The persistent unmixed regions tend to be red regions at the junctions of fused drops. This raises the possibility that nucleosome density affects chromatin dynamics in condensates, due to nucleosome–nucleosome interactions, or nucleosome–Mini-Ph interactions. We conclude that although a fraction of Mini-Ph in Mini-Ph–chromatin condensates is mobile, the chromatin polymers mix slowly and incompletely, a process that could maintain partial compartmentalization of Mini-Ph-bound chromatin over short timescales.

**Ph SAM, but not its polymerization activity, is required for formation of phase-separated condensates**. To test whether Ph SAM is important for condensate formation by Mini-Ph, we prepared Mini-Ph lacking the SAM (Mini-PhΔSAM), or lacking the HD1/FCS domains (Mini-PhΔFCS) (Fig. 3a; Supplementary Fig. 1A). The structure of Ph SAM, including its two polymerization interfaces, termed end helix (EH) and mid loop (ML), is well

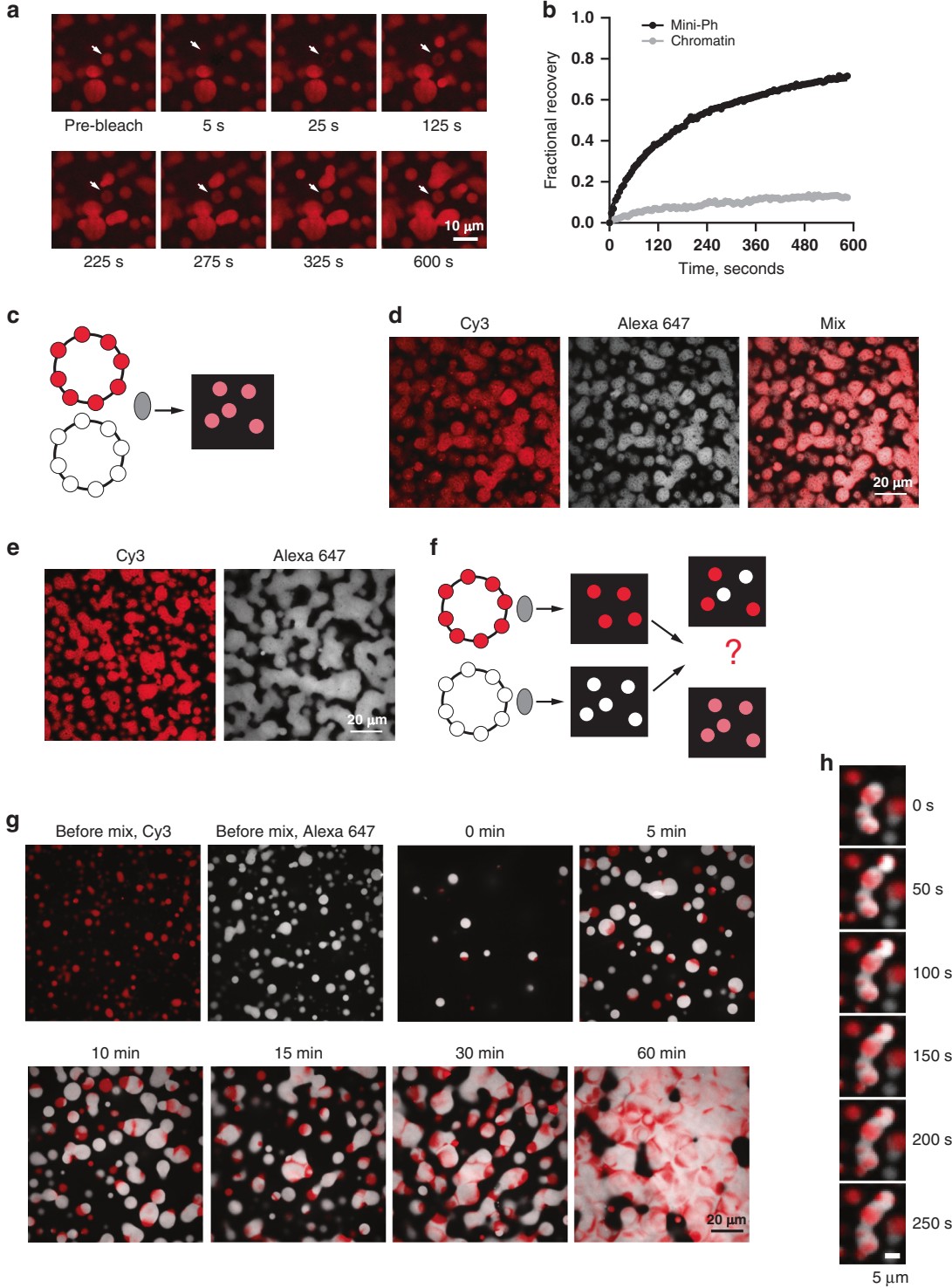

**Fig. 2 Mini-Ph-chromatin condensates intermix slowly in vitro although Mini-Ph is dynamic. a** FRAP experiment demonstrating that Mini-Ph-chromatin condensates exchange Mini-Ph. The structure indicated with the arrow was bleached at $t = 0$. **b** Representative FRAP traces for Mini-Ph or chromatin (after 28 or 27 min of condensate formation, respectively). Data were fit with a double-exponential function (Eq. (1)). For Mini-Ph, % fast = 25; $T1/2_{Fast} = 30$ s; $T1/2_{Slow} = 199$ s; mobile fraction (plateau) = 0.79. For chromatin, % fast = 19; $T1/2_{Fast} = 21$ s; $T1/2_{Slow} = 289$ s; mobile fraction = 0.14. Data are representative of three independent experiments. **c** Scheme for mixing chromatin labeled with different colors before adding Mini-Ph. **d** Mini-Ph-chromatin condensates formed with an equal mix of Cy3 and Alexa-647-labeled chromatin. **e** Mini-Ph-chromatin condensates formed with either Cy3 or Alexa-647-labeled chromatin. **f** Scheme for mixing Mini-Ph-chromatin condensates formed with differently labeled chromatins. **g** Mini-Ph-chromatin condensates were formed with Cy3- or Alexa-647-labeled chromatin and mixed together. The images shown are representative of three independent experiments. **h** Time lapse of fusion of condensates in the mixing experiment. See also Supplementary Fig. 6.

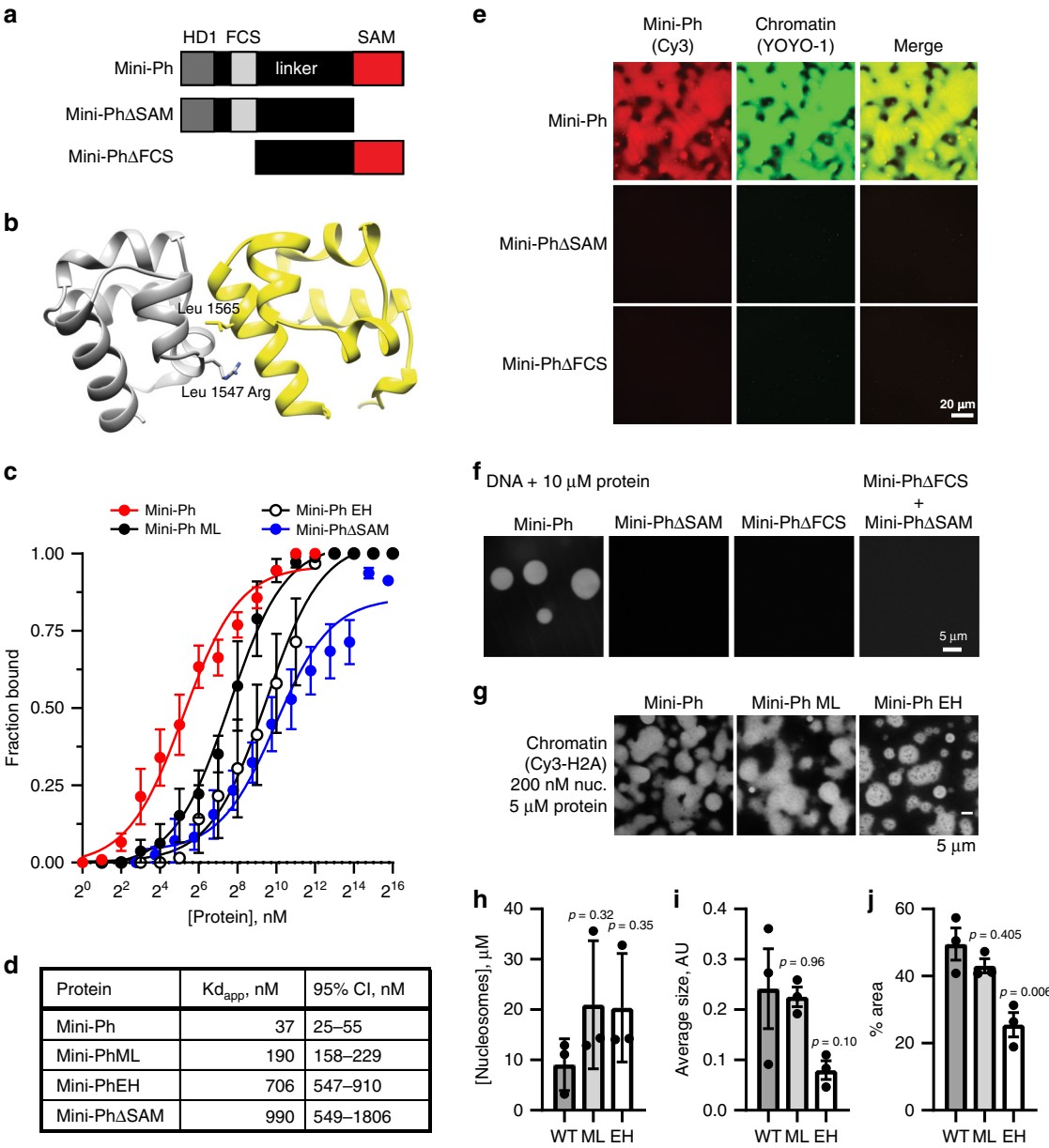

**Fig. 3 Ph SAM, but not its polymerization activity, is essential for condensate formation in vitro. a** Schematic diagram of Mini-Ph truncations. **b** Structure of the Ph SAM–SAM interface indicating the position of the ML and EH mutations that impair SAM polymerization. The EH mutation (Leu 1565 Arg) has a stronger effect on polymerization than the single ML mutation (Leu 1547 Arg). The figure was prepared from the structure of the ML mutant, PDB 1D 1KW4. **c** Summary of filter-binding experiments to measure DNA binding. Points show the mean ± SEM (independent experiments: Mini-Ph $n = 10$; Mini-Ph-ML $n = 3$; Mini-Ph EH $n = 5$; Mini-Ph∆SAM $n = 4$). **d** Kd$_{app}$ with 95% CI for each protein, calculated from the data shown in C using Eq. (2). **e** Both the SAM and the FCS/HD1 region are required for formation of phase-separated condensates with chromatin. **f** Both the SAM and the FCS/HD1 region are required for formation of phase-separated condensates with DNA. **g** Representative images of condensates formed by Mini-Ph or the polymerization mutants (ML and EH) in the presence of chromatin (1-h incubation). **h** The concentration of nucleosomes in condensates formed by wild-type (WT) and polymerization mutants (ML and EH) is similar. **i, j** EH forms smaller condensates with chromatin than WT Mini-Ph, as determined by measuring the average size of the condensates (I, not significant), or the % area covered by condensates (**j**). p Values for **h**–**j** are for one-way ANOVA comparing each sample to the WT control, with Dunnett's correction for multiple comparisons. Bars show the mean ± SEM of three experiments; nine images were analyzed for each experiment. See also Supplementary Fig. 7.

characterized[22] (Fig. 3b). Mutation of these interfaces blocks SAM polymerization in vitro and impairs Ph function in vivo[5,22,24]. We therefore prepared Mini-Ph containing a point mutation that disrupts the EH interface (L1565R) (Mini-Ph EH), or a single point mutation that weakens but does not fully disrupt the ML interface (L1547R) (Mini-Ph-ML) (Supplementary Fig. 1A). Previous AUC experiments with these mutants indicate that Mini-Ph-ML forms shorter polymers than Mini-Ph, and Mini-Ph EH is largely

monomeric, at most forming some dimers at high concentrations (see Fig. 3 of ref. [5]) We first measured the DNA-binding activity of each of these proteins using double-filter binding with a 150-bp DNA probe (Fig. 3c, d; Supplementary Fig. 7A). Mini-Ph binds DNA with an apparent Kd (Kd$_{app}$) of 37 (95% CI: 25–55) nM (calculated with Eq. (2)). Partial disruption of polymerization activity with the single ML mutation increases the Kd$_{app}$ to 190 (95% CI: 158–229) nM. The more severe EH mutation further

increases the Kd$_{app}$ to 706 (95% CI: 547–910) nM, similar to the Kd$_{app}$ of Mini-PhΔSAM (990 (95% CI: 549–1806) nM). DNA binding was not detected with Mini-PhΔFCS by filter binding or EMSA (Supplementary Fig. 7B), indicating that Ph SAM does not bind DNA. Consistent with this conclusion, the Kd$_{app}$ of Mini-PhΔSAM is similar to that for Mini-Ph EH. The much lower Kd$_{app}$ of Mini-Ph presumably reflects cooperative binding by Mini-Ph oligomers. We do not know what the oligomeric state of Mini-Ph is at the concentrations where DNA binding is observed. The Kd$_{app}$ of the SAM–SAM interaction was previously measured as ~200 nM using an immobilized SAM[22], but it is possible that Mini-Ph oligomerization occurs at lower concentrations, which would be consistent with the observed high-affinity binding. We conclude that the polymerization activity of Ph SAM increases the affinity of Mini-Ph for DNA.

Neither Mini-PhΔSAM nor Mini-PhΔFCS forms condensates with chromatin or with DNA (Fig. 3e, f). A mixture of the two proteins also does not form condensates with DNA (Fig. 3f). Thus, both the SAM and the HD1/FCS domains are required for phase separation. We then tested the Mini-Ph polymerization mutants (Mini-Ph-ML and Mini-Ph EH). We find that both form phase-separated condensates with chromatin or DNA under the same conditions as Mini-Ph (Fig. 3g, Supplementary Fig. 8). While the concentration of nucleosomes in condensates is similar (Fig. 3h), condensates formed with Mini-Ph EH are smaller (Fig. 3i, j).

To look more carefully at the effects of the Ph SAM mutations, we titrated Mini-Ph EH or Mini-Ph-ML with DNA over a range of NaCl concentrations, and scored each reaction as one- or two-phase (Supplementary Fig. 8). We find that both mutants are more sensitive to NaCl than Mini-Ph (Supplementary Figs. 3A, B and 8A–C). ATP has been shown to dissolve many protein–RNA condensates, and is hypothesized to have a physiological role in regulating phase separation[55]. To test whether ATP might also regulate Mini-Ph-chromatin condensates, we formed condensates with Mini-Ph, Mini-Ph-ML, or Mini-Ph EH, and challenged them with 2 mM ATP for 15 or 60 min (Supplementary Fig. 9A). Condensates are smaller after ATP treatment, and Mini-Ph EH is more sensitive than either Mini-Ph or Mini-Ph-ML (Supplementary Fig. 9B–E). Treatment of Mini-Ph condensates with 8 mM ATP completely dissolves them (Supplementary Fig. 9F). We conclude that the Ph SAM, and the HD1/FCS regions are both required for condensate formation, while Ph SAM polymerization activity, which increases DNA-binding affinity and changes the oligomeric state of Mini-Ph, enhances condensate formation but is not required for it. Although this result may seem surprising, it is consistent with Mini-Ph existing in a limited oligomeric state prior to condensate formation that cannot be increased further.

Mini-Ph EH and Mini-PhΔSAM have similar DNA-binding activities (Fig. 3c), but different abilities to form condensates (Fig. 3d–f). This indicates that the SAM imparts an activity (presumably protein–protein interactions) that is distinct from the effect on DNA binding and polymerization, but essential for condensate formation. The unstructured linker that connects the FCS/HD1 to the Ph SAM (Supplementary Fig. 4A) was previously shown to interact with the SAM (in trans) by nuclear magnetic resonance[5]. This linker–SAM interaction may allow homotypic interactions between Mini-Ph molecules, even when SAM–SAM interactions are disrupted (as in Mini-Ph EH) and contribute to phase separation. It is also possible that weak SAM–SAM interactions can occur in the EH mutant[5] and contribute to phase separation. Ph SAM polymerization may thus indirectly contribute to phase separation by clustering the DNA-binding FCS domains (increasing multivalency) and increasing the affinity for DNA/chromatin. Supplementary Fig. 10 summarizes

the known and hypothesized interactions that may underlay phase separation by Mini-Ph and DNA or chromatin.

**DNA binding and phase separation modify lysine accessibility in Mini-Ph.** To explore how Mini-Ph interactions change on formation of phase-separated condensates, and how SAM polymerization affects them, we used a mass spectrometry-based protein-footprinting method to probe accessible lysines in Mini-Ph (Fig. 4a). We incubated Mini-Ph alone, or with three different amounts of DNA. In the 1× DNA condition (1 Mini-Ph per 10 bp) and 2× DNA conditions, phase-separated condensates form, while increasing the DNA amount to 16× prevents their formation (Fig. 4b). To display the data, we generated a heat map of the average accessibility at each lysine under each condition using Eq. 3 (Fig. 4c). To compare these values, we used two-sided student's t tests at each lysine position. Accessibility of lysines in the HD1 and FCS domains of Mini-Ph is changed on binding DNA: K1302 and K1340 of HD1 in Mini-Ph are less accessible in the 2× DNA conditions (condensates present), while K1298 and K1302 are less accessible in the 16× DNA condition (Fig. 4c). As the ratio of DNA to Mini-Ph increases, the accessibility of three lysines in the FCS domain (K1370, K1376, and K1380) decreases relative to Mini-Ph alone (Fig. 4c). These decreases in accessibility are consistent with this region being protected by binding to DNA, and indeed, K816 of PHC1 (equivalent to K1380 in Ph) was previously identified as a nucleic acid-binding residue[47] (Supplementary Fig. 12A, B). Changes in accessibility could also reflect changes in protein conformation, particularly in HD1, which is not known to bind DNA. The accessibility of the three lysines in the linker region is low, and does not significantly change with addition of DNA. This is consistent with the linker being in a collapsed state (Supplementary Fig. 4), although the low number of lysines in the linker limits the resolution of the analysis. The accessibility of lysines in the SAM is low both with and without DNA, with no significant changes (Fig. 4c).

To validate global changes in accessibility, we also compared average accessibility of all lysines in each domain under the different conditions (Supplementary Fig. 11C). This confirms the reduction in accessibility of the HD1 and FCS under conditions where condensates form, and no change in the accessibility of the SAM (Supplementary Fig. 11C). These data are consistent with the SAM maintaining its folded structure and pre-existing polymeric state on binding DNA and in condensates. Prolonged incubation in sulfo-NHS acetate leads to dissolution of condensates (Supplementary Fig. 12A–C), likely by disrupting binding of Mini-Ph to DNA. Indeed, if Mini-Ph is fully acetylated with sulfo-NHS acetate, it does not bind DNA, and does not form condensates with DNA (Supplementary Fig. 12D–F).

We then compared accessibility of lysines in Mini-Ph to that in Mini-Ph EH, which does not form polymers. The pattern of lysine accessibility in Mini-Ph EH is distinct from that of Mini-Ph, and differences are not restricted to the SAM (Fig. 4d). Three lysines in the HD1, one in the FCS, and one in the SAM, are significantly altered in Mini-Ph EH vs. Mini-Ph. When differences are considered over each domain, they are more striking (Fig. 4e). While the overall accessibility of the HD1 is the same between the two, probably because both increases and decreases in accessibility are observed, the FCS is less accessible in Mini-Ph EH than in Mini-Ph, while the linker and the SAM are more accessible (Fig. 4d–f). The accessibility of the SAM is consistent with the expected monomeric state of Mini-Ph EH and the positions of the lysines in the SAM polymer structure (Fig. 4f). However, the changes in the other domains of Mini-Ph EH indicate that SAM polymerization likely affects the whole conformation of Mini-Ph and the interactions available for phase

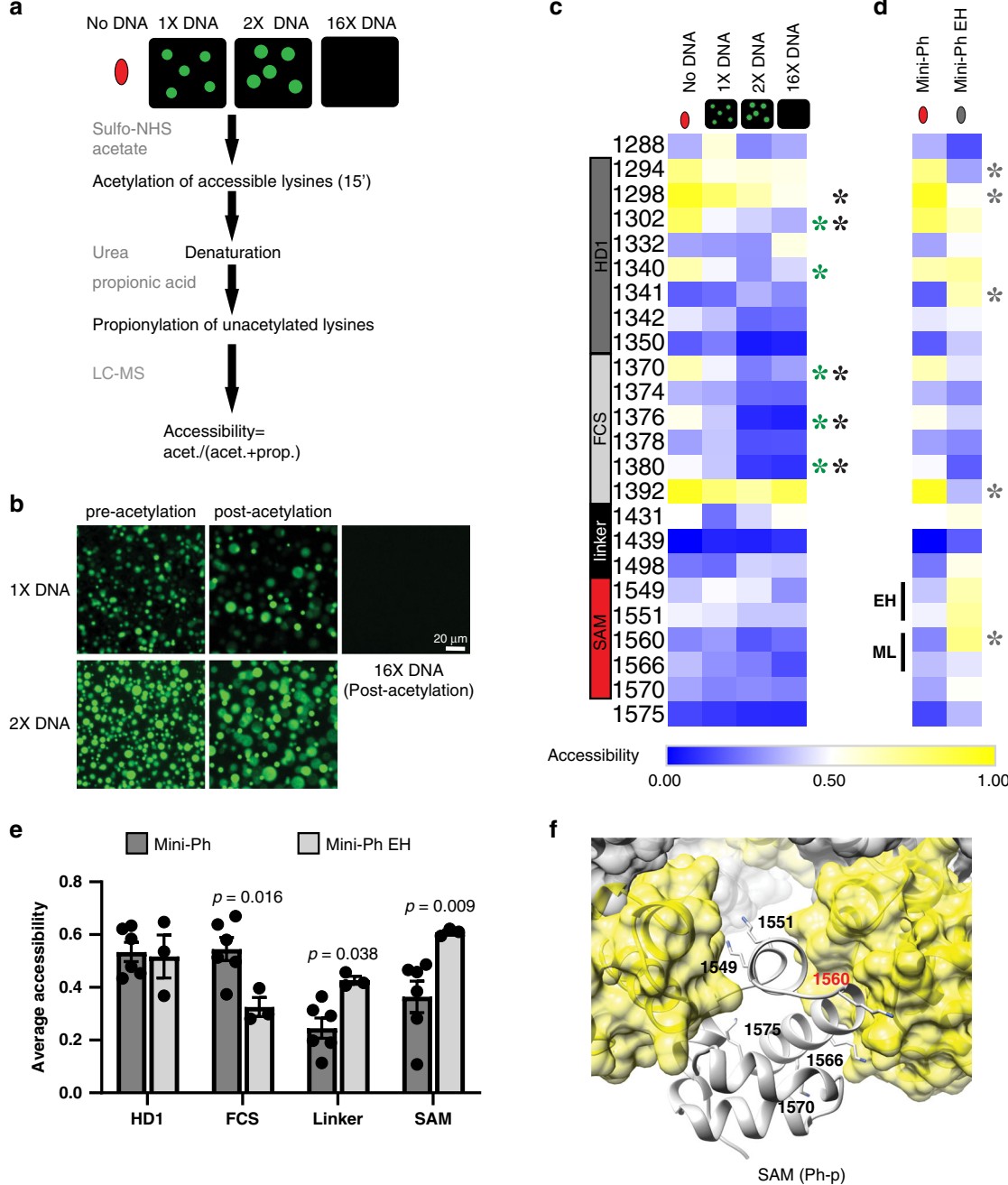

**Fig. 4 Lysine accessibility in Mini-Ph-DNA condensates. a** Schematic of lysine-footprinting assay. Mini-Ph or Mini-Ph EH alone, or in the presence of 1×, 2×, or 16× DNA, is treated with sulfo-NHS acetate to acetylate-accessible lysines. The protein is denatured and treated with propionic acid to propionylate unacetylated lysines. Samples are processed for mass spectrometry, and accessibility is quantified as fraction acetylated for each lysine position (Eq. (3)). **b** Mini-Ph-DNA condensates before and after 15 min of acetylation reaction. **c** Heat map showing accessibility for Mini-Ph alone or with the indicated DNA amounts (Mini-Ph alone, 2× DNA, 16× DNA, $n = 6$; 1× DNA, $n = 3$ independent experiments). **d** Heat map comparing lysine accessibility in Mini-Ph vs. Mini-Ph EH (Mini-Ph, $n = 6$; Mini-Ph EH, $n = 3$ independent experiments). Heat maps are not scaled so that accessibility can be compared across rows and columns. Color scale is from blue (low accessibility) to yellow (high accessibility; the range is 0–1). Asterisks indicate significant differences between samples with and without DNA by two-tailed student's $t$ test with Holm–Sidak correction for multiple comparisons (green = 2× DNA vs. no DNA, black = 16× DNA vs. no DNA, and gray = Mini-Ph vs. Mini-Ph EH). **e** Average accessibility of lysines in each Mini-Ph region compared between Mini-Ph and Mini-Ph EH. Accessibility of all residues in each region was averaged for each replicate and the averages compared across conditions by two-tailed student't $t$ test with Holm–Sidak correction for multiple comparisons. $n$ Values are as stated in (**c**, **d**). Error bars are SEM. **f** Structure of the Ph-p SAM polymer (PDB 1D 1KW4) with lysine side chains shown and labeled for the central SAM unit. Red highlights the residue with significantly changed accessibility in Mini-Ph vs. Mini-Ph EH. Structural data are not available for the HD1 residues studied in the footprinting assay. See also Supplementary Figs. 11–13.

separation. The changes in the HD1 both on condensate formation and between Mini-Ph and Mini-Ph EH also raise the possibility that this domain contributes interactions to phase separation, which will need to be directly tested. Whether Ph SAM would also affect the conformation of Ph in the context of the full-length protein, or when it is in PRC1 (an interaction mediated by HD1) remains to be determined. Finally, we attempted to analyze lysine accessibility in Mini-Ph EH condensates (Supplementary Fig. 13), but the condensates dissolved within 5 min of adding the acetylation reagent. About 5 min of acetylation results in most of the protein being inaccessible (Supplementary Fig. 13C, D). After 15 min of acetylation, accessibility is similar with and without DNA (Supplementary Fig. 13E), consistent with DNA binding being completely disrupted. Comparison of Mini-Ph EH alone after 5 min or 15 min of acetylation indicates that lysines in the HD1 and FCS domains may be more accessible than the linker and SAM (Supplementary Fig. 13C, F). This is consistent with SAM–SAM and/or linker–SAM interactions (Supplementary Fig. 10), although other explanations are possible.

**Ph SAM polymerization affects the mobility of Mini-Ph in condensates**. The experiments presented above indicate that Ph SAM polymerization increases the DNA-binding affinity of Mini-Ph (Fig. 3c), increases the driving forces for phase separation (Fig. 3f–j, Supplementary Figs. 8 and 9), and changes the accessibility of Mini-Ph (Fig. 4). To determine whether the polymeric state of Mini-Ph also affects the material properties of condensates, we compared Mini-Ph and Mini-Ph EH mobility in condensates formed with chromatin (Supplementary Fig. 14). In side-by-side experiments, recovery of fluorescence is consistently faster with Mini-Ph EH (Supplementary Fig. 14A, C, D). We fit FRAP data to a double-exponential function (Eq. (1)). The T1/2 for both slow populations is lower for Mini-Ph EH, the % of molecules in the fast fraction is higher for Mini-Ph EH, and the mobile fractions are similar for both (Supplementary Fig. 14G–J). To analyze chromatin mobility, we analyzed the Cy3 label on H2A in the same condensates used to collect FRAP traces for Alexa-647-labeled Mini-Ph or Mini-Ph EH before and after bleaching (Supplementary Fig. 14B, E, F). Less than 10% of the fluorescence is recovered over the 5-min experiment for condensates formed with Mini-Ph and Mini-Ph EH (Supplementary Fig. 13B). Thus, consistent with Fig. 2, chromatin and Mini-Ph or Mini-Ph EH have distinct kinetics in condensates. The slow kinetics of chromatin may be intrinsic to the template since the EH mutation in Mini-Ph does not affect them. We conclude that assembly of Mini-Ph into polymers not only increases the driving force for phase separation, but influences the material properties of the condensates that are formed.

**Mini-Ph–chromatin condensates recruit PRC1 from nuclear extracts**. One function of phase separation is to create biochemical compartments that are enriched for specific components, and can stimulate or inhibit biochemical reactions[26]. To determine whether Mini-Ph–chromatin condensates can create unique biochemical compartments, we asked whether condensates can recruit proteins from nuclear extracts (Fig. 5a). We prepared nuclear extracts from *Drosophila* S2R+ cells, and used an anion-exchange resin to deplete nucleic acids from the extracts. Even after depletion, the nuclear extracts contain substantial amounts of RNA (Supplementary Fig. 15A). Treatment of extracts with RNAseA resulted in precipitation of most of the protein from the extracts, so that we used extracts containing RNA for our experiments. Chromatin alone forms a few tiny structures in extracts (Fig. 5b, c, reaction 1). Mini-Ph does not form condensates in buffer (e.g., Fig. 1c), but does form small condensates in extracts, likely by binding to RNA,

since the condensates stain with YOYO-1 (Fig. 5b, c, reaction 2). When Mini-Ph is incubated with chromatin to form condensates, and then nuclear extracts are added, the condensates are preserved, although they are smaller than condensates in equivalent reactions incubated in buffer (Fig. 5b, c, compare reactions 3 and 4). Although the condensates are smaller, the concentration of chromatin in them is similar to that in condensates incubated in buffer (Fig. 5d). We do not know why the condensates are smaller after incubation in nuclear extracts. Post-translational modifications can influence phase separation[42], but the small-molecule substrates needed for enzymes that mediate them should be depleted in our desalted extracts. The presence of nucleic acids in the extracts could disrupt condensates, analogous to what is observed at high concentrations of DNA (Supplementary Fig. 2F, G). Alternatively, proteins in the extracts that bind to Mini-Ph and/or chromatin may disrupt interactions required for condensates.

We used low-speed centrifugation to isolate condensates (2 min @ 2500*g*) and analyzed their nucleic acid content on agarose gels. When Mini-Ph is incubated with extracts in the absence of chromatin, the pelleted condensates contain RNA (Fig. 5e). When Mini-Ph is incubated with chromatin, and the extract added subsequently, the isolated condensates contain both chromatin and RNA (Fig. 5e). Since the amount of RNA that is pelleted with Mini-Ph is similar with and without chromatin, we infer that Mini-Ph condensates in extracts can contain both RNA and chromatin (Fig. 5e–g). To confirm this, we analyzed colocalization of fluorescently labeled Mini-Ph with chromatin after incubation in buffer, or in nuclear extracts (Supplementary Fig. 15B–D). Most Mini-Ph-containing structures also contain chromatin. This is consistent with chromatin condensates recruiting RNA from the extracts, rather than formation of a separate class of Mini-Ph–RNA condensates.

To analyze the protein components of Mini-Ph condensates in nuclear extracts, we used Western blotting. PRC1 components are enriched in condensates formed in extracts with or without chromatin, while the PRC2 subunits Su(Z)12 and p55, the single-strand DNA-binding protein RPA70, and the chromatin-remodeling complex subunit ACF1 are not enriched (Fig. 5h, i). Thus, Mini-Ph condensates can concentrate endogenous PRC1 provided by nuclear extracts.

**Mini-Ph-chromatin condensates enhance ubiquitylation of histone H2A**. To determine whether the PRC1 recruited to Mini-Ph condensates is active, we tested whether chromatin present in condensates is ubiquitylated on histone H2A. When extracts were supplied with ATP and ubiquitin, very low levels of ubiquitylated H2A (H2A-Ub) were detected. Addition of the E1 ubiquitin-activating enzyme and E2 ubiquitin-conjugating enzyme along with ATP and ubiquitin resulted in detectable H2A-Ub in extracts (Fig. 6a, b). Formation of Mini-Ph–chromatin condensates prior to incubation in extracts increased H2A-Ub by about twofold. This suggests that PRC1 recruited to condensates is functional, and that Mini-Ph–chromatin condensates enhance the ubiquitylation reaction (Fig. 6b, c).

To determine if the Ph SAM polymerization state can influence condensate formation in the more physiological environment of nuclear extracts, we prepared condensates with Mini-Ph-ML or Mini-Ph EH, and added nuclear extracts to them. Mini-Ph-ML condensates behave similar to those formed with Mini-Ph in extracts (Supplementary Fig. 16). In contrast, incubation of Mini-Ph EH condensates in extracts transforms them into diffuse structures that occupy a larger area but have a reduced chromatin concentration relative to condensates incubated in buffer (Supplementary Fig. 16). We tested histone ubiquitylation in extracts in the presence of Mini-Ph-ML or Mini-Ph EH, and find that neither

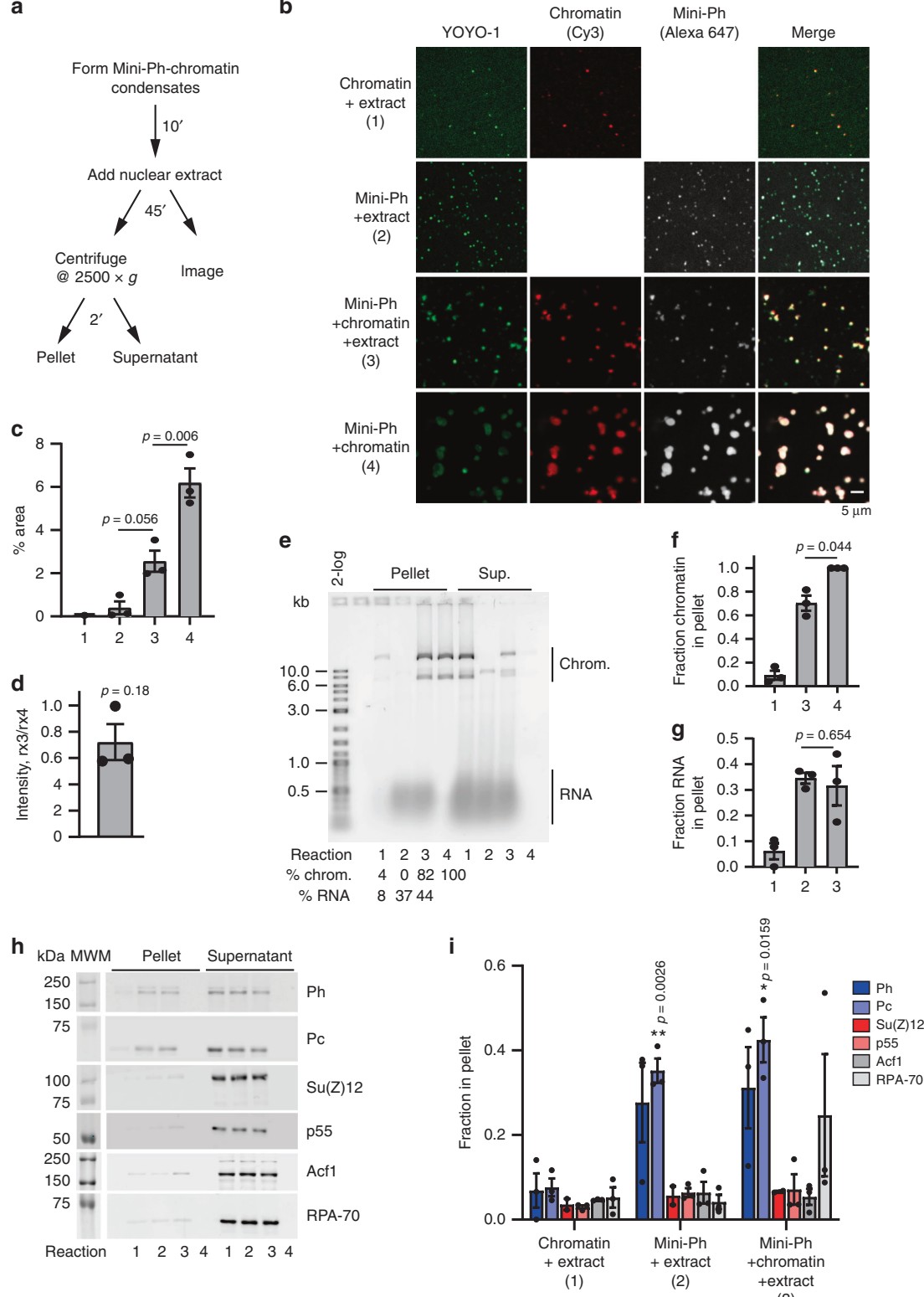

mutant stimulates histone ubiquitylation (Fig. 6b, c). We do not know if this is because the condensates formed by the polymerization mutants have different properties (e.g., Supplementary Fig. 14), or because they recruit less PRC1, as might occur if SAM–SAM interactions (between Mini-Ph and Ph in PRC1) are directly involved in recruiting PRC1 to chromatin.

The observation that Mini-Ph condensates increase histone ubiquitylation might reflect the increased concentration of PRC1

in condensates (Fig. 5h, i). It is not necessarily predicted, however, that the environment of condensates, in which chromatin is compacted, would enhance enzyme activity. Thus, to determine whether Mini-Ph-chromatin condensates enhance PRC1 activity under optimal conditions, we reconstituted the ubiquitylation reaction in vitro, using chromatin alone or Mini-Ph–chromatin condensates as the substrate (Supplementary Fig. 17). We used PRC1ΔPh for these experiments (Supplementary Fig. 1B), which

**Fig. 5 Mini-Ph condensates recruit PRC1 from extracts. a** Scheme for isolating Mini-Ph-chromatin condensates from nuclear extracts. **b** Representative images of condensates formed in each of the four indicated reactions. **c** Quantification of phase-separated condensates (% area covered by condensates, nine images analyzed for each of three experiments using YOYO-1 staining). p Values are for one-way ANOVA with Tukey's correction for multiple comparisons. **d** Ratio of average intensity in condensates formed by Mini-Ph + chromatin + nuclear extracts (reaction 3) vs. Mini-Ph + chromatin (reaction 4) for 3 experiments. p Value is for one-sample t test comparing the ratio to the expected value of 1. **e** SYBR Gold stained gel of nucleic acid content of pelleted reactions. Reactions 1–4 are as indicated in panel B for (**c–i**). Summary of three experiments quantifying the fraction of chromatin (**f**) and RNA (**g**) in the pellet. p Values are for paired two-tailed t test between reactions 3 and 4. **h** Representative Western blots of one experiment analyzing the content of pelleted condensates. Equal amounts of pellet and supernatants were loaded. **i** Summary of three experiments analyzing the content of condensates formed in extracts. Su(Z)12 was only analyzed in two experiments. One-way ANOVA was used to compare all three samples for each antibody with Tukey's multiple-comparison test. All bar graphs show mean, and error bars are SEM. See also Supplementary Fig. 15.

can interact with Mini-Ph via the HD1 domain (but unlike PRC1 found in extracts, not via SAM–SAM interactions), and is fully active as an E3 ligase. PRC1ΔPh catalyzes formation of H2A-Ub on chromatin in a dose-dependent manner (Fig. 6d; Supplementary Fig. 17C, D). When Mini-Ph–chromatin condensates are used as the substrate, the activity of PRC1ΔPh is increased by about twofold over the entire titration, indicating that condensates stimulate PRC1ΔPh activity (Fig. 6d, e). We also analyzed condensates at the end of the reactions to confirm that they persist under reaction conditions (Fig. 6f, g). Because a high fraction of the histones is ubiquitylated in these experiments (Fig. 6d, e), these results indicate that H2A-Ub does not disrupt condensates.

**Ph SAM affects ubiquitylation of H2A in vivo.** To test whether the activity of Ph SAM is important for histone ubiquitylation in vivo, we used Drosophila S2 cell lines that express Ph or Ph with the double ML mutation (L1547R/H1552R), which disrupts Ph SAM polymerization as effectively as the EH mutant used in our in vitro studies, under control of an inducible promoter[10]. We isolated histones from control S2 cells and cells induced to overexpress Ph or Ph-ML, and measured the levels of H2A-Ub (Fig. 7a–d). Cells overexpressing Ph have an approximately twofold increase in overall H2A-Ub relative to control cells (Fig. 7b). Cells overexpressing Ph-ML have increased H2A-Ub in some experiments, but this difference was not significant, even though Ph-ML is expressed at higher levels than Ph (Fig. 7a, c).

Because we find that Ph SAM polymerization activity is not strictly required for phase separation in vitro, we wondered if Ph-ML might be able to phase-separate in vivo, particularly when present at high concentrations. Formation of highly concentrated foci in cells is consistent with phase separation, although it can arise through other mechanisms, as has been pointed out[37]. To test whether Ph-ML can form foci in cells, we transiently transfected Drosophila S2 cells with Venus-tagged Ph, Ph-ML, or PhΔSAM under control of the heat-shock promoter. After heat-shock induction, Venus-Ph forms large, round, bright foci. These foci are mainly (although not exclusively) nuclear, and little Venus signal is observed in the nucleoplasm outside the foci (Fig. 7e). In contrast, Venus-PhΔSAM is uniformly distributed in the nucleus, and does not form foci (Fig. 7f). Venus-Ph-ML forms foci but is also distributed throughout the nucleus (Fig. 7g). Thus, foci formation in vivo and phase separation in vitro are correlated with each other and with enhanced histone ubiquitylation. We tested Venus-Mini-Ph in Drosophila S2R + cells, and find that, unlike Venus-Ph, it does not form foci in most cells. In about 7% of the cells, it forms a single focus, which can be quite large (Supplementary Fig. 18A, B, D); these unusual foci are not observed with Venus-Mini-PhΔSAM (Supplementary Fig. 18C, D). Thus, although Ph SAM is required for foci formation in cells, the other disordered regions of Ph shape its behavior in cells as has been observed for other proteins that can undergo LLPS[42].

## Discussion

We have identified phase separation as a new activity of the Ph SAM, the domain that is most clearly implicated in large-scale chromatin organization by PcG proteins. Our data are consistent with two possible functions of Ph SAM-dependent phase separation: (1) formation of a compacted chromatin state with dynamic components and (2) creating a unique biochemical compartment that enhances PRC1-mediated histone modification.

In developing Drosophila embryos, Ph lacking the SAM cannot rescue any Ph functions, while Ph with a polymerization interface mutated can partially rescue Ph function, although with defects in transcriptional repression[24]. In vitro, Mini-Ph lacking the SAM does not form phase-separated condensates, while Mini-Ph with the polymerization interface mutated (Mini-Ph EH) does form condensates although they are smaller. In Drosophila tissue culture cells, Ph lacking the SAM does not form foci, while polymerization-defective Ph (Ph-ML) can form foci when over-expressed (Fig. 7). Thus, foci formation in vivo, and phase separation in vitro, are correlated with full Ph function, and the LLPS activity of Ph SAM may be the critical function of the SAM that remains even when polymerization is disrupted.

Previous work implicates Ph polymerization in both transcription repression and chromatin organization[5,10,11,24,25]. Analysis of PcG proteins in normal Drosophila tissue culture cells or those that mildly overexpress Ph or Ph-ML using stochastic optical reconstruction microcsopy (STORM) showed that normal Drosophila tissue culture cells contain hundreds of nanoscale clusters, although only a few large PcG bodies are visible by conventional microscopy[10]. Mild overexpression of Ph increased the number but not the size of clusters, and increased long-range contacts, while overexpression of the strong Ph-ML mutant disrupted clusters and reduced long-range contacts. This work and work in mammalian cells[11] directly implicates Ph SAM polymerization in the nanoscale organization of PcG proteins and large-scale organization of chromatin.

Although Ph SAM alone can form open-ended polymers, the extent to which long SAM polymers occur in the context of the full protein is unclear. In vitro, the oligomeric state of Mini-Ph is limited to four to six units[5]; this can be explained by the action of the unstructured linker that separates Ph SAM from the FCS in conjunction with the helical configuration of SAM polymers[22]. Steric considerations suggest that Ph SAM polymerization may be even further restricted in the context of PRC1. Thus, the contribution of polymerization to LLPS may be much subtler than would occur with an actual open-ended Ph SAM polymer. The linker connecting the SAM to the FCS is not conserved in Ph homologs (Supplementary Fig. 4). The linker of PHC3, unlike the Drosophila Ph linker[5], does not bind the PHC3 SAM in trans[5], and allows much more extensive SAM polymerization than that of Ph[5]. It is therefore possible that the linker has been tuned across evolution to control polymerization and its interplay with phase separation. This is consistent with modeling-based analysis, indicating that the properties of linkers connecting interacting

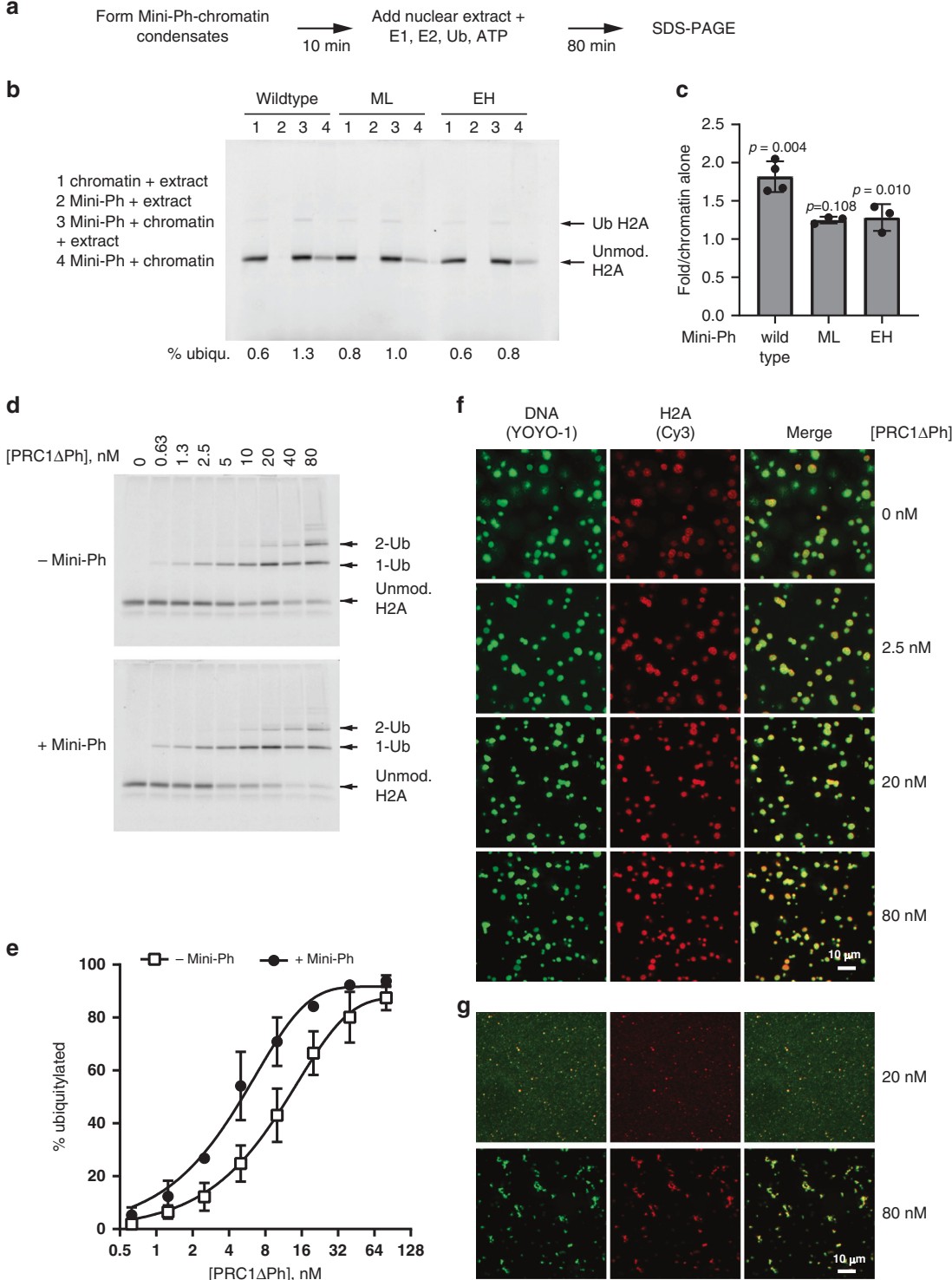

**Fig. 6 Mini-Ph condensates facilitate histone ubiquitylation by PRC1. a** Scheme for carrying out ubiquitylation assays in nuclear extracts. **b** Representative SDS-PAGE of Cy3-labeled H2A showing ubiquitylation of chromatin in nuclear extracts in the presence or absence of Mini-Ph (wild type), Mini-Ph-ML, or Mini-Ph EH. Note that the condensates formed in buffer (reaction 4) were poorly recovered in this experiment. **c** Quantification of three independent ubiquitylation assays. Bars are mean ± SEM. *p* Values are for one-sample *t* test comparing values to the expected value of 1. **d** Representative SDS-PAGE of Cy3-labeled H2A showing ubiquitylation reaction with PRC1ΔPh and chromatin in the presence or absence of Mini-Ph. **e** Quantification of ubiquitylation reactions, *n* = 3. Points are the mean ± SEM and data were fit with an exponential function. **f** Microscopy of Mini-Ph-chromatin condensates at the end of fully reconstituted in vitro ubiquitylation reactions. **g** Fibrous condensates are formed by high concentrations of PRC1ΔPh in the absence of Mini-Ph; at lower concentrations, no structures are visible. Imaging was conducted for all three independent ubiquitylation experiments and the results were similar. See also Supplementary Figs. 16 and 17.

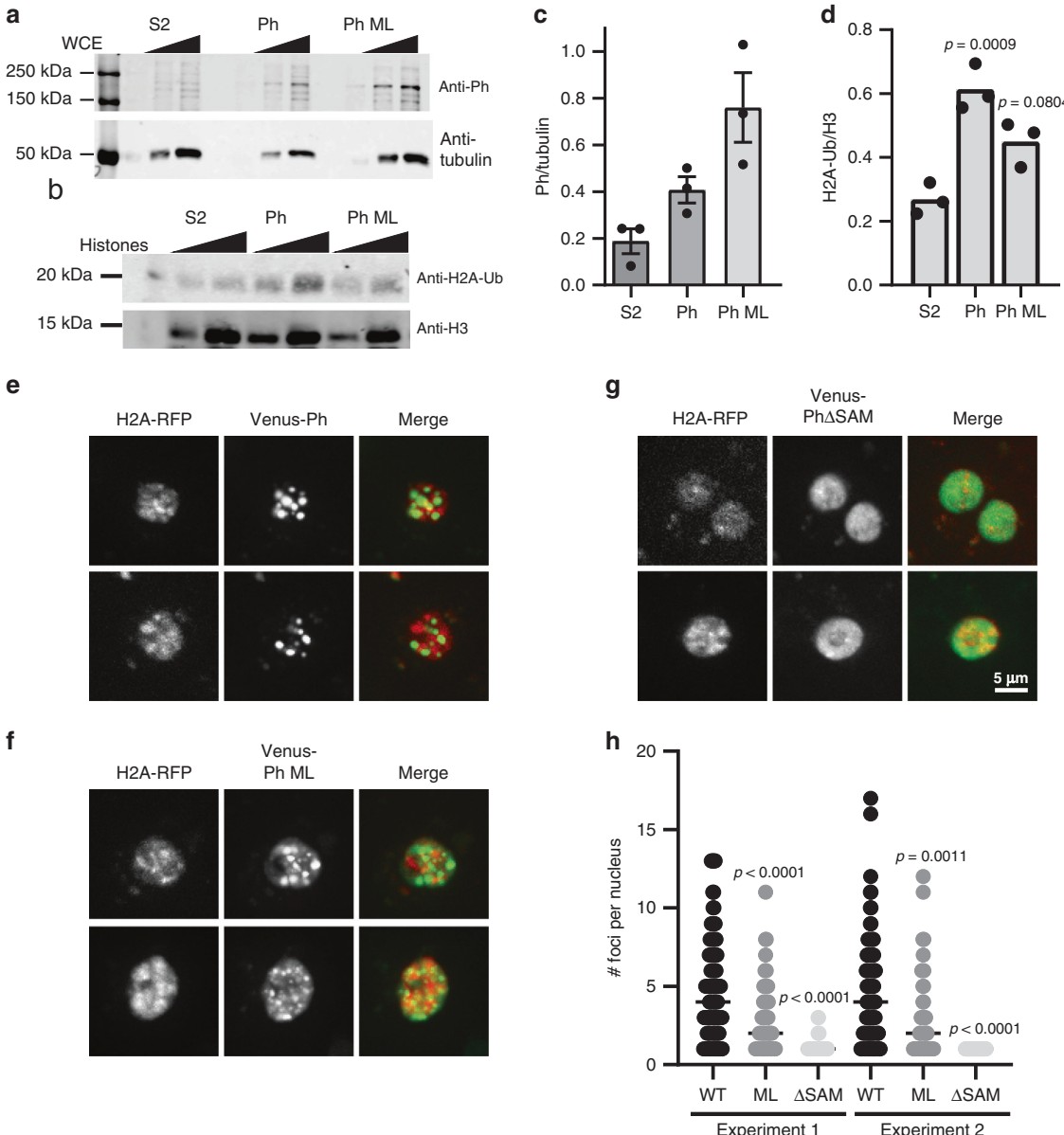

**Fig. 7 Ph with an intact SAM increases H2A-ub in vivo. a** Representative Western blot of Ph levels in induced cell lines. Note that Ph-ML is the strong double mutant (L1547R/H1552R), which was previously shown to disrupt Ph clustering in cells[10]. **b** Representative Western blot of histone H2A-Ub levels in induced cell lines. Blots were reprobed with anti-H3 to normalize loading. In total, 125 and 250 ng of acid-extracted histones were loaded for each sample. **c, d** Quantification of Ph (**c**) and H2A-Ub (**d**) for three experiments. *p* Values are for one-way ANOVA comparing Ph-WT and Ph-ML cells to control (S2) for the 250 ng point. **e–g** Representative images of cells overexpressing Venus-Ph (**e**), Venus-Ph-ML (**f**), or Venus-PhΔSAM (**g**) by transient transfection under control of the heat-shock promoter. Scale bar shown in (**g**) applies to (**e–g**). **h** Graph of the number of foci per cell for two independent experiments. Note that only cells with >zero foci were included. *p* Values are for Kruskal–Wallis tests with Dunn's multiple-comparison correction. Comparisons were made among cells expressing different versions of Ph for both experiments. *n* Values (# cells analyzed in experiments 1 and 2) in Venus-Ph 118, 144; Venus-Ph-ML 128, 63; Venus-PhΔSAM 72, 30. See also Supplementary Fig. 18.

domains tune phase-separation properties[28]. Two other PcG proteins, SCM and Sfmbt, also have SAMs, and the three SAMS have been shown to co-assemble[56]; joining of SAM-mediated polymers of these three proteins could allow formation of large and diverse polymers. Evaluating the phase-separation activity of these other PcG SAMs, alone or in combination, and of Ph homologs, will be an important future goal.

The phase-separation activity of Ph SAM is also likely subject to negative regulation. A disordered, serine-/threonine-rich sequence adjacent to HD1 undergoes O-linked glycosylation mediated by the PcG protein Sxc[24,57]. This region and Sxc are both important for Ph function in regulation of some genes[24,57]. In the absence of glycosylation, Ph undergoes SAM-dependent "non-productive aggregation," which is not alleviated by mutating the Ph SAM polymerization interfaces[24]. It is possible that "non-productive aggregation" in fact reflects SAM-dependent phase separation (or maturation of phase-separated protein into stable, insoluble aggregates)[26]. The glycosylated sequence is not part of Mini-Ph. Mini-Ph is produced in *E. coli*, and is not glycosylated, yet Mini-Ph is soluble. It therefore seems likely that the effect of glycosylation, although dependent on Ph SAM, also involves other sequences in Ph. We speculate that the glycosylated region may restrict Ph SAM-mediated phase separation, and preliminary in vitro data support this idea (E.S and N.J.F, unpublished observation).

A hallmark of LLPS is that it depends on weak, multivalent interactions that allow rapid reorganization and unrestricted stoichiometry. The polymerization activity of Ph SAM may contribute multivalent interactions. However, additional interactions are required to cross-link SAM-mediated polymers[58], which (at least in Mini-Ph) involve HD1 and/or the FCS. Based on the comparison between Mini-Ph and Mini-Ph EH, linker–SAM and (possibly) SAM–SAM interactions that do not require an intact polymerization interface likely also contribute (Supplementary Fig. 10). In vitro, dynamic SAM polymerization is not likely to directly drive phase separation by Mini-Ph because the Kd for polymerization is so much lower than the saturation concentration at which phase separation occurs. However, in the polymerization mutants, and in vivo where the concentration of Ph is lower[51,59], dynamic polymerization of Ph SAM could control phase separation. In LLPS of Mini-Ph with chromatin or DNA, the role of the FCS is likely nucleic acid binding; however, HD1 and/or the FCS may form additional protein–protein interactions (Fig. 4). It is interesting to note that Sfmbt and SCM, the other two SAM-containing proteins also contain an FCS, although the distance and additional motifs separating the FCS from the SAM varies. The combination of an FCS (i.e., a nucleic acid-binding domain) and a SAM could allow these proteins also to undergo phase separation. In support of this idea, the *Caenorhabditis elegans* SOP-2 protein functions as a PcG protein[60], and forms large nuclear bodies[32]. Although it is not a clear sequence homolog of Ph, SOP-2 consists of an RNA-binding motif, an intrinsically disordered region (IDR), and a SAM[32]. Recently, the IDR of SOP-2 was shown to undergo LLPS in vitro, induced by crowding agents or RNA[42]. Addition of the SAM to the IDR still allowed LLPS, but resulted in formation of smaller condensates that showed lower recovery in FRAP experiments[42].

A model for the function of Ph SAM that can reconcile the seemingly different requirements for the SAM and its polymerization activity in different contexts is that Ph SAM drives at least three different states. First, Ph SAM polymerization activity may drive formation of tiny PcG clusters that mediate local repression of transcription simply through cooperative binding interactions. This is consistent with our finding that Ph SAM and its polymerization activity increases the DNA-binding affinity of Mini-Ph, at concentrations well below the range where phase separation occurs (Fig. 3). It is also consistent with the dependence of Ph-repressive activity when targeted to a reporter gene on Ph SAM polymerization activity[5]. Second, bridging of nucleosomes mediated by the polymerization interfaces of Ph SAM associated with chromatin-bound PRC1 may drive collapse of the chromatin polymer over larger regions of PRC1-bound chromatin[10,45,61,62]. Indeed, a model of this process could explain the observed effects of overexpressing Ph with the strong ML mutation or wild-type Ph, which increases the number but not size of Ph clusters[10]. In cases where the local concentration of Ph is very high, Ph may undergo LLPS mediated by multivalent interactions among Ph molecules and between Ph and chromatin (or Ph and RNA), as captured by our in vitro assays, and possibly in the foci observed when Venus-Ph is overexpressed in cells (Fig. 7). Which mechanism dominates in any situation could be modulated by the local concentration of PcG proteins (i.e., how strong a PcG recruitment site is, or the density of recruitment sites). This could be analogous to the distinction between enhancers and superenhancers, which recruit higher levels of transcription factors and cofactors, and where LLPS is believed to occur[36,63]. There is also no reason at this time to exclude hybrid models[53]. For example, LLPS could be a mechanism to create biochemical compartments, and within these domains, strict SAM–SAM interactions could establish precise chromatin contacts required for gene repression. LLPS may also represent an extreme and transient state, used to silence large chromatin domains rapidly during development[18,64], or as a step in re-establishing gene expression patterns during the cell cycle. All of these possibilities remain to be tested, but the separation of phase separation and polymerization activity revealed by our simple in vitro assays may provide a means to do so.

Many proteins with diverse localizations and functions have SAMs. Some SAMs have been shown to polymerize in a concentration-dependent manner, while others require additional recruitment mechanisms to induce polymerization. The SAMS of a subset of proteins, including Ets1, Fli1, and p63[65], has not been observed to polymerize. It is therefore possible that phase separation is a property of the SAM that is distinct from polymerization, a hypothesis that is testable by measuring the phase-separation activity of proteins with monomeric SAMs.

We find that Ph SAM-driven chromatin condensates can enhance PRC1-mediated histone ubiquitylation. We do not know what the mechanism of stimulation of H2A-Ub is. It is unlikely to be concentration of the reaction components in condensates because all of the components (except PRC1ΔPh) are present at saturating concentrations in these reactions. PRC1ΔPh binds chromatin tightly (Kd for 150-bp DNA is ≤1 nM[66]) so that Mini-Ph is also not needed to recruit PRC1ΔPh to chromatin. Although further experiments will be needed to determine the mechanism, the environment of condensates may stimulate steps in the reaction subsequent to substrate binding, which could include the actual ubiquitin transfer or steps affecting processivity[67]. It has recently been shown that H2A-Ub mediated by PRC1 is stimulated by chromatin compaction[68], and that spreading of H2B-Ub along chromatin is facilitated by formation of structured, phase-separated compartments by the ubiquitylation machinery[69], which may be relevant to our observations. Formation of protein–chromatin condensates with the heterochromatin protein HP1 alters the conformation of the nucleosome, rendering specific regions of the histone proteins more accessible[70]. It is possible that nucleosome conformation is also changed in Mini-Ph condensates, and that these changes facilitate histone ubiquitylation. Detailed characterization of chromatin in condensates will be an important future goal.

Stimulation of H2A-Ub is unlikely to be the essential function of the Ph SAM in *Drosophila*, since the modification is not required for PRC1-dependent gene repression in vivo, including repression of genes that depend on Ph SAM[71,72]. However, H2A-Ub is required for full development[71,72]. *Drosophila* cPRC1 also does not seem to mediate most H2A-Ub in tissue culture cells, and it is likely that another ncPRC1 containing L3(73)Ah, a homolog of mammalian Pcgf3, in place of PSC, is present in these cells[73]. This also means that in our experiments with nuclear extracts, although we observe PRC1 recruitment to condensates, we cannot be certain that it is responsible for the ubiquitylation activity we observe (Fig. 6).

Histone ubiquitylation by PRC1 has been most intensively studied in mouse embryonic stem cells (mESCs), where systematic analysis of the effect of disrupting PRC1 subunits implicates ncPRC1 (i.e., non PHC-containing) in creation of most H2A-Ub[12–15]. However, using an artificial tethering system that allows PcG proteins to be reversibly targeted to a reporter gene so that persistent effects on chromatin and gene expression (i.e., memory) can be measured, Moussa et al.[74] found that heritable gene repression and propagation of H2A-Ub depend on cPRC1. Recent work indicates a central role for H2A-Ub in PcG-dependent gene regulation in mESCs[12–14], in seeming contrast with observations in *Drosophila*; it will be interesting to determine how Ph SAM contributes to H2A-Ub activity in mammals. The ability of Ph SAM to condense chromatin and to promote H2A-Ub could be important for rapidly building PcG chromatin

domains, or restoring them at the end of mitosis. H2A-Ub is not detected on mitotic chromosomes in mammalian cells[75,76], suggesting that it is reacquired after cells exit mitosis.

Finally, Cbx2, a member of some mammalian canonical (PHC-containing) PRC1s, which has a strong chromatin-compacting activity[77], has also been shown to form phase-separated condensates with chromatin in vitro, and to form 1,6-hexanediol-sensitive foci in ES cells[41,43]. This phase- separation activity is mediated by a charged IDR in Cbx2 that is important for the developmental function of Cbx2[78]. Further, as shown in Supplementary Fig. 17, Mini-Ph does not form foci in cells, indicating that other sequences in Ph, all of which are predicted to be disordered, can regulate the activity of the Ph SAM. How the activity of Ph SAM is regulated by other sequences in Ph and coordinated with that of other components of PRC1, particularly that of PSC which has a powerful chromatin-compacting activity analogous to that of Cbx2[79], is an important question for future study.

## Methods

**Cloning**. Cloning of Mini-Ph and the polymerization mutants was described previously[5]. Mini-PhΔSAM (residues 1291–1507) and Mini-PhΔFCS (residues 1397–1577) were cloned into a modified pET-3c vector expressing a leader sequence containing a hexahistidine tag followed by a TEV cleavage site. To express Venus-tagged proteins in S2 cells, Ph, Ph-ML, or PhΔSAM were first cloned into a house-modified gateway donor vector and full sequences confirmed. Gateway LR Clonase II (Thermo Fisher) was used to perform LR recombination with pHVW from the DGRC (stock # 1089) to produce the final expression plasmids.

**Protein purification**. His-tagged Mini-Ph, Mini-Ph-EH, and Mini-Ph-ML were expressed in Rosetta (DE3) *E. coli*. Cultures were grown at 37 °C to an OD of 0.8–1.0, and then shifted to 15 °C for overnight induction with 1 mM IPTG. Cells were pelleted, flash-frozen, and stored at −80 °C. Cells were resuspended in 2 ml/g lysis buffer (50 mM Tris, pH 8.5, 200 mM NaCl, 10 mM β-ME, 100 μM ZnCl₂, 0.2 PMSF, and 0.5 mM benzamidine). Cells were incubated on ice for 10 min, flash-frozen in liquid nitrogen, thawed at 37 °C, and sonicated 6*30 s at 30% intensity. Freeze–thaw and sonication were repeated, and the lysate centrifuged for 1 h at 100,000*g and 4 °C. Cleared lysate was sonicated 6*30″ at 40% intensity, and filtered through a 22-μm filter. Lysate (from 1 L) was applied to a 1-ml His-Trap (all FPLC columns were obtained from GE Healthcare) column using an AKTA FPLC, and eluted with a gradient of imidazole (from 10 to 300 mM) in lysis buffer. Fractions with Mini-Ph were dialyzed overnight against 1 L of 20 mM Tris, pH 8.5, 50 mM NaCl, 100 μM ZnCl₂, and 10 mM β-ME. Dialyzed fractions were centrifuged for 10 min at 20,800*g, and loaded on a 1-ml HiTrapQ-HP column and eluted with a gradient from 50 mM to 1 M NaCl in binding buffer. Fractions were pooled and dialyzed overnight into 20 mM Tris, pH 8, 50 mM NaCl, 10 μM ZnCl₂, and 1 mM βME, aliquotted, and stored at −80°C. For one of the three preparations used, Mini-Ph was further purified by size-exclusion chromatography using a Superose 12 column.

Mini-PhΔSAM and Mini-PhΔFCS proteins were expressed in BL21 (DE3) Gold cells pretransformed with the pRARE plasmid. The transformed cells were grown at 37 °C in LB media to an OD₆₀₀ of ~0.7–0.8 and induced overnight at 15 °C. Cells harvested from 1 L of culture were resuspended with 10 ml of lysis buffer (50 mM Tris, pH 8.0, 200 mM NaCl, 5 mM βME, 30 mM imidazole, pH 7.5, and 1 mM PMSF) and lysed by sonication. The soluble lysates were introduced onto an Ni-NTA column, washed with lysis buffer (without PMSF), and bound proteins eluted using 300 mM imidazole, 200 mM NaCl, and 5 mM βME. The leader sequence was cleaved using TEV protease, and the cleaved sequence and uncleaved proteins removed by passing through a Ni-NTA column. Further purification was performed using a HiTrapQ-HP column. Fractions containing protein were pooled, buffer-exchanged into 50 mM Tris, pH 8.0, 100 mM NaCl, and 5 mM βME, and concentrated. Mini-PhΔSAM was further purified on a Superdex 200 size-exclusion column in 50 mM Tris, pH 8.0, 100 mM NaCl, and 5 mM βME. Purified, concentrated proteins were stored at −80 °C.

The following plasmids were used to prepare the human ubiquitylation machinery: human 6×-His-UBA1 (E1) (pET21d-Ube1, addgene #34965), Human UbcH5c (E2) (pET28a-UbcH5c, addgene # 12643), and 6×His-Ubiquitin (pET15b-His-Ub) (kind gift of B. Schulman). Proteins were expressed in *E. coli* and purified essentially as described[80,81]. His-Ube1 was purified by Ni-NTA affinity followed by Superdex 200 chromatography[80]. UbcH5c was purified on a HiTrap SP-XL column followed by Superdex 200[81]. 6×-His-Ub was purified by Ni-NTA chromatography.

*Xenopus laevis* histones, including H2B-122C mutant, were expressed in and purified from *E. coli*, using standard protocols[82,83]. Histones were expressed individually, and purified from inclusion bodies using Q sepharose (to remove nucleic acids) followed by SP sepharose under denaturing conditions. Histones were dialyzed against H₂O and lyophilized. All experiments were carried out with

histone H3 with Cys110 (the only cysteine natively present in the histones) mutated to Ala.

Fluorescent labeling of histone H2A with NHS-Cy3 was carried out under low pH conditions favoring labeling of the N-terminal amine. Lyophilized H2A was resuspended in labeling buffer (20 mM Hepes, pH 6.2, 7 M Guanidium HCl, and 5 mM EDTA) to a concentration of 0.1 mM. NHS-Cy3 stock (in dimethyl formamide) was added to a final ratio of 0.5:1 (dye to histone) and incubated at room temperature for 90 min. Free dye was removed with Amicon concentrators, after diluting with labeling buffer without Guanidium to reduce the Gu-HCl concentration to 6 M. In some cases, Zeba spin columns (Thermo Fisher, 7MWCO) were used instead to remove free dye. To label H2B-122C with maleimide-Alexa 647, lyophilized histone was reconstituted in denaturing labeling buffer (20 mM Tris-HCl, pH 7.0, 7 M guanidium HCl, and 5 mM EDTA) to a final concentration of 0.1 mM followed by treatment with a 100-fold excess of TCEP for 30 min. Maleimide-Alexa 647 was added to a final ratio of 3:1 (dye:histone) and incubated for 3 h at room temperature. The labeling reaction was quenched with β-ME (final concentration 80 mM), and free dye removed as above. Octamer reconstitutions and purification on a Superdex 200 size-exclusion column were carried out as described[82,83]. Briefly, lyophilized histones were resuspended in unfolding buffer (20 mM Tris-HCl, pH 7.5, 7 M guanidium HCl, and 10 mM DTT) and mixed at a molar ratio of 1 H3, 1 H4, 1.2 H2A, and 1.2 H2B. When labeled histones were used, they were mixed with the unlabeled histones in labeling buffer. The mixture of histone subunits was adjusted to 1 mg/ml and dialyzed against 3 changes of octamer-refolding buffer (2 M NaCl, 10 mM Tris, pH 7.5, 1 mM EDTA, and 5 mM β-ME). Octamers were then concentrated and applied to a Superdex 200 size-exclusion column in octamer-refolding buffer. Octamer-containing fractions were pooled and dialyzed against octamer- refolding buffer containing 50% glycerol and stored at −80 °C.

PRC1ΔPh was purified from nuclear extracts of Sf9 cells infected with baculoviruses for the three subunits (Flag-PSC, Pc, and dRING) for 3 days[66]. Standard nuclear extracts were prepared, except that nuclei were purified through a sucrose cushion prior to nuclear extraction exactly as described[84]. We find that this step reduces co-purification of tubulin. During the purification, the 2 M KCl wash in the published protocol was replaced with a wash consisting of BC2000N + 1 M urea (20 mM Hepes, pH 7.9, 2 0.4 mM EDTA, 2 M KCl, 1 M deionized urea, and 0.05% NP40, no glycerol). Additionally, prior to eluting the protein, anti-FLAG beads were incubated with 3–5 volumes of BC300N (20 mM Hepes, pH 7.9, 300 mM KCl, 0.2 mM EDTA, 20% glycerol, and 0.05% NP40) with 4 mM ATP + 4 mM MgCl₂ for 30 min at room temperature. This step reduces the amount of HSC-70 that copurifies with PRC1ΔPh. Protein was eluted with 0.4 mg/ml FLAG in BC300 without NP40, concentrated to ~1 mg/ml, and stored at −80 °C in BC300N.

**Fluorescent labeling and acetylation of Mini-Ph and other proteins**. To fluorescently label proteins, NHS-ester-Cy3 or Alexa-647 were used to randomly label lysines. A Zeba column was used to buffer-exchange the protein into 20 mM Hepes, pH 7.9, 200 mM NaCl for Mini-Ph, or BC300N for proteins expressed in Sf9 cells (to remove Flag peptide used to elute the proteins); labeling was carried out with a 0.5:1 (dye:protein) ratio for 15 min at room temperature. Labeling was quenched by addition of lysine to 10 mM. Free dye was removed using two Zeba columns, which were equilibrated in the labeling buffer. Labeled protein was mixed with unlabeled at a ratio of between 1:10 and 1:25, depending on the labeling efficiency, for imaging experiments. Acetylation of Mini-Ph was carried out exactly as for fluorescent labeling, except that a ratio of 8:1 sulfo-NHS-acetate:lysine residues in Mini-Ph was used and labeling was carried out for 1 h at room temperature.

**Preparation of nuclear extracts from *Drosophila* S2R+ cells**. S2R+ cells (Drosophila Genome Research Center) were grown in M3-BYPE media (Sigma) with 10% fetal bovine serum (FBS) (Weisent). In total, 20*15-cm dishes were used to prepare nuclear extracts as described[85], except that nuclei were purified through a sucrose cushion prior to extraction. Cells lysed in hypotonic buffer were layered over two volumes of 30% sucrose in hypotonic buffer, and centrifuged 18′ @ 1400g. Nuclei were washed once in hypotonic buffer, and extracted as described. The high- salt-extraction buffer was 1.2 M KCl, and extracts were not dialyzed. To use the extracts to treat condensates, up to 100 μl of extract was buffer-exchanged into 20 mM Tris, pH 8, 50 mM NaCl using a Zeba column. Extracts were centrifuged 2′ @ 20,000g and incubated for 15′ on ice with 60% volume of Q sepharose. Extracts were spun through an empty column (2′ @ 10,000g), and then centrifuged 2′ @20,000g. All procedures were carried out on ice or at 4 °C and contained protease inhibitors and 0.4× PhosStop (Sigma) phosphatase inhibitor.

**Chromatin preparation**. Most experiments were carried out with the plasmid p5S*8, which contains 5 blocks of 8–5S nucleosome-positioning sequences (repeat length 208 base pairs). Plasmids were assembled by salt-gradient dialysis as described[86]. Chromatin was finally dialyzed into HEN (10 mM Hepes, pH 7.9, 0.25 mM EDTA, and 10 mM NaCl) buffer and stored at 4 °C. To measure chromatin assembly, 100 ng of each assembly was digested overnight with 10 U of EcoRI in NEB buffer 2.1, and loaded on a 0.5× TBE, 5% acrylamide native gel. Gels

were stained with Ethidium bromide and imaged on a Typhoon imager. For quantification, the nucleosomal signal is multiplied by 2.5 to account for the quenching effect of bound protein on Ethidium bromide[87]. For micrococcal nuclease analysis, 800–1000 ng of chromatin was diluted into 40 μl of the following buffer: 12 mM Hepes, pH 7.9, 0.12 mM EDTA, 60 mM KCl, and 2 mM MgCl₂ and split into 4 tubes. Micrococcal nuclease (Sigma, #N3755) (0.5 U/μl in 50 mM Tris, pH 8, 0.05 mM CaCl₂, and 50% glycerol) was diluted 1:18, 1:54, 1:162, and 1:486 in MNase dilution buffer (50 mM Tris, pH 8.0, 10 mM NaCl, 126 mM CaCl₂, and 5% glycerol). 1 μl of each dilution was used to digest chromatin for 7 min at room temperature. Reactions were stopped with DSB-PK (10× stock: 50 mM Tris, pH 8.0, 0.1 M EDTA, 1% sodium dodecyl sulfate (SDS), and 25% glycerol + 10 mg/ml Proteinase K), digested overnight at 50 °C, and analyzed on 1× TBE–1.5% agarose (SeaKem) gels that were stained with Ethidium bromide and imaged on a Typhoon Imager.

**Phase-separation assays**. Proteins and templates were routinely centrifuged full speed in a microfuge for 2–5 min at 4 °C to remove aggregates before setting up phase-separation assays. For phase-separation assays, reactions (10–20 μl) were assembled in a 384-well glass-bottom imaging dish (SensoPlate, Greiner Bio-One). Wells were not pretreated; precoating with bovine serum albumin (BSA) did not influence phase separation by Mini-Ph. Phase separation was initiated by addition of the protein or the DNA, and mixing the reaction by gently pipetting up and down three times, with care taken not to introduce air. Reactions were incubated in the dark for 15 min or up to several hours. For reactions where YOYO-1 (Thermo Fisher) was used, it was added at the beginning of the reaction to a final dilution of 1:3000. Typical reaction conditions are 50 mM NaCl or 50 mM KCl, 20 mM Tris, pH 8.0. Reactions were set up on ice, and transferred to room temperature for 15 min. Turbidity measurements were made in duplicate using a NanoDrop spectrophotometer. Phase-separated condensates were pelleted by centrifugation at 14,000g for 2 min at 4 °C, and supernatants removed to fresh tubes. Pellets were resuspended in 12 μl of 1.5× SDS-sample buffer, and 6× SDS-sample buffer was added to the supernatant. About 10% of the pellet and supernatant were removed and digested in DSB-PK for 2 h at 50 °C for DNA analysis. The remainder of the sample was boiled and analyzed by sodium dodecyl sulfate polyacrylamide gel electrophoresis (SDS-PAGE).

**Imaging of condensates**. All images were collected on a Zeiss microscope, equipped with a Yokogawa CSU-1 spinning-disk confocal head. Zen 2012 software was used for image acquisition with a 63× oil objective, or a 100× oil objective (for movies and FRAP) and evolved EMCCD camera from Photometrics. The excitation wavelengths for YOYO/Venus, Cy3/RFP, and Alexa 647 were 488, 561, and 639 nm, respectively.

**Measuring nucleosome concentration in condensates**. Images were collected at 25% laser power, 200-ms exposure for buffer, chromatin alone, a titration of labeled histone octamers (in octamer-refolding buffer, which contains 2 M NaCl, and in which histone octamers remain assembled), and Mini-Ph-chromatin condensates. Histones are the same histones used to prepare chromatin; 43% of the histone octamers are labeled (measured both using the NanoDrop and by loading histones and free dye on SDS-PAGE gels), corresponding to a 21.5% labeling efficiency on H2A (since there are two copies of H2A in each octamer). Image J measure was used to measure the mean gray intensity for each of 9 images for each point. Images were manually checked and images with bright artifacts removed, although these had little impact on the measured intensities. A linear regression was fit to the calibration curve and used to convert measured intensities to nucleosome concentrations. To measure intensities in condensates, Image J was used to threshold the images (AutoThreshold-->Li); Analyze Particles was used to measure the mean gray intensity in each thresholded structure. Particle size was set as 100-infinity pixels. The mean gray intensity from the buffer image was subtracted from all measurements, which were converted to nucleosome concentrations using the calibration curve.

**Fluorescent recovery after photobleaching**. FRAP experiments were carried out with Alexa-647-labeled Mini-Ph or Mini-Ph EH. Bleaching was done with a 595-nm laser, for 1500 ms. This effectively bleaches both Alexa-647 Mini-Ph and Cy3-H2A, although we were only able to record FRAP images from one channel. Two prebleach images were collected, followed by an image every 5 or 10 s. All FRAP analyses of Mini-Ph were done by bleaching single complete structures. Images were analyzed in Image J (Fiji). An ROI was selected for the bleach area, background, and a nonbleached structure. Background-subtracted, normalized data were fit with a double-exponential fit (Eq. (1)) using GraphPad Prism 8.

$$Y = Y0 + \text{SpanFast} * \left(1 - e^{-\text{KFast} * X}\right) + \text{SpanSlow} * \left(1 - e^{-\text{KSlow} * X}\right). \quad (1)$$

We excluded data sets that could not be fit, and obvious technical artifacts (e.g., if a drop fuses with the bleached condensate during the experiment).

**Image analysis of condensates**. Images for display were prepared using Zen2 (blue edition). For quantification, images were exported as TIFs from Zen (original

data). Image J (Fiji) was used to threshold the images (Li algorithm); thresholds were manually checked and images with too few structures to threshold were removed. Areas and intensities of thresholded structures were measured using Image J (Analyze Particles, size = 10-infinity pixels). For colocalization analysis, the GDSC-- > Colocalization-- > Particle Overlap was used. Masks were created in the Alexa 647 (Mini-Ph) and Cy3 (chromatin) channels, and overlap of Cy3 with Mini-Ph structures measured.

Movies were created from .czi files in Image J (Fiji). Movies were saved as .avi files at 1, 2, or 3 frames per second, and using PNG compression. Movies were subsequently converted to .mp4 files using Movavi Video Converter 20.

**Filter binding**. Filter binding was carried out as described[66,88]. Briefly, a 150-bp internally labeled DNA probe was prepared by PCR and gel-purified. The probe was used at 0.02 nM. Reaction conditions were 60 mM KCl, 12 mM Hepes, pH 7.9, 0.24 mM EDTA, and 4% glycerol, in a 20-μl volume. Proteins were centrifuged for 2 min at full speed in a microfuge before preparing the dilution series. Binding reactions were incubated for 1 h at room temperature. Hybond-XL was used as the bottom membrane (binds DNA), and was pre-equilibrated in 0.4 M Tris, pH 8.0. Nitrocellulose was used as the top membrane (binds protein + DNA), and was pretreated with 0.4 M KOH for 10 min, neutralized by washing through several changes of Milli-Q water, and equilibrated for at least 1 h in binding buffer. Filters were assembled in a 48-well slot-blot apparatus, and each well washed with 100 μl of binding buffer. The vacuum was turned off, and reactions loaded on the filters. Slots were immediately washed with 2 × 100 μl of binding buffer. Filters were air-dried, exposed to a phosphoimager screen, and scanned on a Typhoon Imager (GE Healthcare). ImageQuant was used to quantify top (bound) and bottom (unbound) filters, and fraction bound calculated in Excel. Curve fitting was done in GraphPad Prism 8, using Eq. (2)

$$Y = AB\_\text{max} * \frac{X}{X + K_d} + b. \quad (2)$$

**Protein-footprinting assay**. The acetylation-footprinting assay is described in detail in Kang et al.[89]. Phase-separation reactions were directly scaled up to use 4 μg of protein for each sample. Condensates were allowed to form at room temperature for 15 min; an aliquot of each sample was removed to confirm phase separation using microscopy. Sulfo-NHS acetate was dissolved immediately before use, and added to a final concentration of 0.5 mM. An aliquot of each sample was removed to monitor phase separation by microscopy, and reactions were stopped after 15 min by addition of trifluoroacetic acid to a final concentration of 1%. For Mini-Ph EH, acetylation of condensates was restricted to 5 min because these condensates dissolved rapidly on exposure to Sulfo-NHS acetate. We therefore analyzed Mini-Ph EH alone, and bound to DNA (16× DNA, Fig. 4) after both 5 and 15 min of acetylation. Samples were TCA-precipitated, denatured with 8 M urea, reduced with DTT (45 mM final concentration), treated with a final concentration of 10 mM iodoacetamide, and diluted 1:2 with H₂O before treating with propionic anhydride twice. Samples were dried, treated with propionic anhydride again, dried, resuspended, and digested sequentially with trypsin and chymotrypsin. Samples were purified with a ZipTip and analyzed by LC–MS/MS on an Orbitrap-Fusion mass spectrometer.

Mass spectrometry data[90] were analyzed using Maxquant (v1.6.10.43) with Acetyl(K) and Propionylation(K) as variable modifications. In total, ten missed cleavages were allowed since lysine modification will block trypsin digest. All data files were analyzed together, with the match between runs option. The intensities for identified Acetyl and Propionyl sites were used for quantification. Accessibility was calculated for each site (in Excel) using Eq. (3)

$$\text{Accessibility} = \left(\text{intensity}_\text{acetylated}\right) / \left(\text{intensity}_\text{acetylated} + \text{intensity}_\text{prop} + 0.5\right). \quad (3)$$

To compare accessibility between samples, GraphPad Prism 8 was used to conduct student's t test, assuming equal variance across samples, and with the Holm–Sidak method of correction for multiple comparisons, with alpha = 0.05 (unpaired, two-tailed test). Heat maps were prepared from averaged accessibilities using Morpheus (https://software.broadinstitute.org/morpheus).

**Analysis of condensates after incubation in nuclear extracts**. Phase-separation reactions were set up in 40 μl with 80 nM nucleosomes, 7.5 μM Mini-Ph, in 20 mM Tris, pH 8.0, and 50 mM NaCl. After incubating for 10 min at room temperature, 12 μl of nuclear extracts were added, and reactions mixed by gently pipetting up and down. About 7.5 μl were removed and diluted to 10 μl for imaging, and 7.5 μl mixed with the uibiquitylation machinery to assay histone ubiquitylation. After 60 min of total incubation, samples were pelleted by centrifugation for 2 min at 2500g, 4 °C. Supernatants were removed and SDS–sample buffer added to 1×. Pellets were resuspended in 2× SDS–sample buffer. About 2 μl of each pellet and supernatant were removed and digested with Proteinase K for at least 1 h at 55 °C before analysis on 1.2% agarose, 1× TAE gels, which were stained with SYBR Gold to visualize nucleic acids. The remainder of the samples were boiled and loaded on 8% SDS-PAGE gels, transferred to nitrocellulose, and used for Western blotting. Membranes were blocked with 5% nonfat dry milk in PBST (PBS + 0.3%

Tween-20), and incubated with primary antibodies diluted in 5% milk-PBST overnight at 4 °C. Membranes were washed 3*10 min in PBST, incubated in secondary antibody diluted in 5% milk-PBST for 1–2 h, washed 3*10 min in PBST, and visualized using a Li-Cor Odyssey imaging system. Image J (Fiji) or Image-QuantTL was used to quantify band intensities. Primary antibodies are as follows: anti-Ph (Rb) (prepared in Francis lab) 1:2000; anti-Pc (Rb) and anti-Su(Z)12 (Rb (gifts of J. Mueller)) 1:5000, 1:3000; anti-Acf1 (Rb) (gift of D. Fyodorov), 1:1000; anti-RPA (Rb) (gift of P. Fisher), 1:3000; anti-p55 (Abcam), 1:1000.

**Histone ubiquitylation assays**. For ubiquitylation assays, 125 ng of chromatin per 5 µl was preincubated with 5 µM Mini-Ph (or buffer) for 15 min at room temperature to induce phase separation, followed by addition of the ubiquitylation machinery and PRC1ΔPh. The final reaction conditions are 40 nM nucleosomes, 20 mM Hepes, pH 7.9, 0.25 mM MgCl$_2$, 0.25 mM ATP, 0.6 mM DTT, 60 mM KCl, 25 mM NaCl, 700 nM E1, 800 nM E2, and 500 ng of Ub. Titrations of the E1, E2, and His-Ub indicate that none are limiting under these conditions. Reactions were further incubated for 45 min at room temperature. Aliquots were removed for imaging, and the remainder of the reaction stopped by addition of SDS–sample buffer. Boiled samples were loaded on 16% SDS-PAGE gels, which were scanned for Cy3 to detect H2A, and then stained with SYPRO Ruby (Lonza). Histone ubiquitylation assays in nuclear extracts were carried out under the same conditions, except that the preincubation of chromatin with Mini-Ph was 10 min, nuclear extracts were added just before the ubiquitylation components, and reactions were incubated for 80 min at room temperature.

**Cell culture**. Wild-type S2 cells (from Expression Systems, 94-005F) and S2 cell lines harboring stable Ph or Ph-ML[10] transgenes were grown in suspension in ESF-921 media (Expression Systems) with 5% FBS. Protein expression was induced with 0.5 µM CuSO$_4$ for 4 days. For whole-cell extracts, cells were resuspended in 2× SDS–sample buffer and boiled. For histone extraction, we followed the protocol of Abcam (https://www.abcam.com/protocols/histone-extraction-protocol-for-western-blot); HDAC inhibitors were not included. Western blots were carried out as described above, except that blots probed with anti-H2A-Ub were blocked with 5% BSA in PBST, and ImageQuant was used to quantify the bands. The antibodies used were anti-Histone H3 (Rb) (Abcam ab 1791) 1:2000; anti-Histone H2A-Ub (Rb) (Cell Signaling Technology, 8240S) 1:1000.

**Live-cell imaging**. For live-cell imaging, S2 (Fig. 7) or S2R+ (Supplementary Fig. 17) cells were plated at 10$^6$ cells per well in 6-well plates the night before transfection. Transfection was carried out using Trans-IT lipid (Mirus), according to the manufacturer's protocol. About 2 µg of each Venus-Ph construct was used along with 0.5 µg of pAct5C-H2A-RFP[91]. One to two days after transfection, cells were replated on ConA-coated imaging dishes (Ibidi). Heat shock was for 8 min (S2R+) or 12 min (S2) at 37 °C, and cells were analyzed within 24 h of protein induction. Confocal stacks of thick slices (3 µm) were collected on the spinning-disk microscope described above using the 63× objective to capture foci throughout the cell.

**Image analysis of live cells**. The .czi files of image stacks were opened in Image J (Fiji), the channels split, and converted to maximum-intensity projections. The red channel (H2A-RFP) was used to segment nuclei as follows. Images were thresholded with the Li algorithm, followed by removing outliers less than 5 pixels, and 3 rounds of erosion. Thresholded images were converted to masks, processed with a watershed algorithm, and Analyze Particles used with a size threshold of 200-infinity pixels to select nuclei. The green channel (Venus fusion proteins) was then processed with "Find maxima" with the following parameters: Prominence: 20000; strict; exclude edge maxima; output type: single points. The nuclei selected from the red channel were used as ROIs, and the # maxima per ROI (i.e., # foci/nucleus) obtained using Measure in the ROI tool, followed by dividing the raw integrated density by 255. This entire pipeline is explained here https://microscopy.duke.edu/guides/count-nuclear-foci-ImageJ. To compare the # foci per cell, cells with zero foci were excluded; since Venus-PhΔSAM does not form foci, the majority of cells were excluded.

**Statistics and reproducibility**. All observations were made in at least three independent experiments, unless otherwise stated (e.g., titration matrices shown in Fig. 1i, Supplementary Figs. 2F, 3A, and 8A, B were conducted twice). Statistics were calculated using GraphPad Prism v8.4.3, using recommended settings, including for correction for multiple comparisons. In cases where $p$ values can be one- or two-tailed, two-tailed values are always reported. Three different preparations of Mini-Ph were tested, with identical results, and two different preparations of PRC1ΔPh. For Mini-Ph EH, Mini-Ph-ML, Mini-PhΔFCS, and Mini-PhΔSAM, a single preparation was used; for the latter two proteins, similar results were obtained with these proteins prepared in Sf9 cells. Multiple preparations of chromatin (greater than 10) were used over the course of these experiments.

**Software used for data collection and analysis**. ZenBlue, Image J Fiji, Image-QuantTL, MaxQuant v1.6.10.43, GraphPad Prism v8.4.3, Excel, Morpheus (https://software.broadinstitute.org/morpheus/), PONDR-VSL2 (http://www.pondr.com/), localCIDER (http://pappulab.github.io/localCIDER/), and Movavi Video Converter 20.

**Reporting summary**. Further information on research design is available in the Nature Research Reporting Summary linked to this article.

## Data availability
Mass spectrometry raw files are available at MassIVE under accession number MSV000085717. All other relevant data supporting the key findings of this study are available within the article and its Supplementary Information files or from the corresponding author upon reasonable request. Source data are provided with this paper. The Source Data file includes measured intensities for ROIs for FRAP traces and fits of these data (Fig. 2, Supplementary Figs. 6, 14), filter-binding data (fraction bound) (Fig. 3c), nucleosome and condensate measurements (Fig. 3h–j), MaxQuant output (intensities) and calculation of accessibility for acetylation-footprinting experiments (Fig. 4), Western blots and quantification (Figs. 5, 7), ubiquitylation assay quantification (Fig. 6), and foci counts (Fig. 7). A reporting summary for this article is available as a Supplementary Information file. Source data are provided with this paper.

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

## Acknowledgements

The authors thank B. Schulman, V. Archambault, and the Drosophila Genomics Resource Center (NIH grant 2P40OD010949) for plasmids, J. Mueller, P. Fisher, and D. Fyodorov for antibodies, E. Lecuyer and J.C. Padilla for S2R+ cells, D. Filion for assistance with imaging experiments, D. Faubert for assistance with mass spectrometry, J. Boulais for assistance in analyzing MS data, and F. Robert for critical reading of the paper. This work was funded by RO1 GM114338-02 to C.A.K. and N.J.F., and CIHR-operating grant 344769 to N.J.F.

## Author contributions

Conceptualization: N.J.F., C.A.K., and E.S.; formal analysis: N.J.F., I.K., and A.K.; investigation: E.S., J.J.K., C.S., A.K., E.L.B., and N.J.F.; resources: C.S., O.S., J.J.K., E.L.B., A.K., and C.A.K.; writing: N.J.F. and C.A.K., with input from E.S.; visualization: E.S., A.K., I.K., N.J.F., and C.A.K.; supervision: N.J.F. and C.A.K.; funding acquisition: C.A.K. and N.J.F.

## Competing interests

The authors declare no competing interests.
