## [Peer Review File · Nature Communications]

REVIEWER COMMENTS

Reviewer #1 (Remarks to the Author):

This paper well-written paper presents a thoughtful analysis of the ability of a key part of the PH protein to generate phase separated structures. PH is a key component of the Polycomb-Group. It is part of the PRC1 complex and has been previously shown to be involved in long range interactions involving its SAM domain. There is, therefore, strong justification for studying its role in phase separation given the potential connection between higher order networking and phase separation.

Strengths of the paper are that the experiments are carefully done and not over-interpreted. Multiple models are thoughtfully considered in the discussion. There is compelling data presented showing that the mini-PH protein can phase separate in vitro to concentrate template, and interesting analysis of mixing of templates (Fig. 2) and a nice demonstration that other PRC1 components can partition into the phase separated droplets. Ubiquitylation studies are used to validate that the PRC1 in droplets is functional, and cell culture studies show that droplets can form in cells. Overall, this is an interesting paper that contributes to the field and is useful in further understanding polycomb phase separation and enzymatic activity and the complexities of multivalent, intramolecular and intermolecular interactions that have a significant role in epigenetics. I saw no significant weaknesses and offer a few comments for consideration, mainly on presentation.

Comments:

- 1) In terms of organisation, there were a few issues with the model that were thoroughly addressed in the discussion but lacked nuance in the results section. Introduction of some of these complexities earlier would be helpful to the reader for clarity purposes (for example, the conflicting views of ncPRC1 and cPRC1 ubiquitylation activity).
- 2) Figures could be more clearly referenced – there were several times the text brought up questions that were then answered separately by a closer look at the figures but this could be mitigated by increased clarity in figure references throughout the results section.
- 3) If complex coacervation is to be discussed, analysis of protein charge should be included in the first section of the results.
- 4) Lysine accessibility is analyzed with respect to mini-PH, but it seems unclear how the rest of the protein would affect this data? In addition, it would be helpful to see mini-PH's behavior in vivo as compared to the full protein? This experiment needs some qualification in the text as to what conclusions can be reached and what caveats there are when thinking about the natural context of these domains.
- 5) It would be interesting to understand the conclusions from 6B and 6C at multiple concentrations of protein (WT or PH mutant input). In addition, how is protein concentration of the recruited extract being normalized between the samples? For example, is ubiquitylation activity in the polymerization mutant equivalent to less concentrated WT sample? This might suggest polymerization ability is linked to recruiting ability? It is not clear what is meant by fragile in this context – are you suggesting that the same amount is recruited but then activity is still lower because interactions are less robust?

Reviewer #2 (Remarks to the Author):

First and foremost, I offer my sincere and unvarnished apologies to the authors for my excessive tardiness in turning this review around. The situation brought on by Covid-19 and the massive ramp up in responsibilities, especially teaching and mentoring, have taken a significant toll on bandwidth. While this is a legitimate excuse, I recognize that it will not and should not assuage the

authors sense of grievance.

In this work, the authors focus on the phase behavior of a truncated version of Polyhomeotic (Mini-Ph) with chromatin. The hypothesis is that the transcriptional repression is organized by the formation of protein-DNA condensates. To test this hypothesis, the authors assess the condensate forming ability of Mini-Ph with chromatin. They query the internal dynamics of molecules within these condensates, the exchange of material across condensates, and the fusion dynamics of condensates. Based on deletion mutagenesis experiments the authors show that the sterile alpha motif, a polymerization domain, is essential for forming condensates, although this does not appear to derive from the polymerization activity of SAM - which is surprising. Further characterizations of lysine acetylation and histone ubiquitylation are provided.

Overall, the experiments are interesting, well designed, and well thought out. The conclusions stay confined to the constraints imposed by the data and the manuscript certainly merits publication following some key revisions listed below.

1. Please provide a summary of the sequence features of the unstructured linker.
2. Please note that Figure 1G is not a phase diagram. I respect the qualifier of this being a qualitative phase diagram, but nevertheless request that this be referred to as a limited coarse-grained delineation of the boundary between the one- and two-phase regimes.
3. To invoke complex coacervation as the mechanism of phase separation, one would have to establish that the central and perhaps only contribution to the driving forces for phase separation are the extent of mismatches between charges on the DNA and protein. This would require a systematic assessment of the effects of charge imbalance by titrating salt and pH in different combinations and investigating how the critical point is crossed as a function of these titrations. That would be too much to ask. The simpler fix is to indicate that the mechanism likely involves a combination of homotypic and heterotypic interactions. If and only if the mechanism is solely driven by complementary electrostatic interactions and the release of condensed counterions can one invoke a purely complex coacervation mechanism.
4. The estimate of intra-condensate concentrations of nucleosomes will be predicated on the standard curve used to calibrate these measurements. How reliable are such calibrations for measuring concentrations in dense phases? The 60x mismatch between calibrations based on free dye vs. the approach used here would appear to suggest that there might be photo physical artifacts. What is unclear is if the current approach does not rely on the use of labeled molecules. Please clarify.
5. Figure 2B should be reanalyzed to quantify the contributions of mobile vs. immobile fractions at different time points. This is necessary since the recovery is not 100%. Further, the traces for chromatin are not shown here. This is important to include in order to support the observations made in the text. Also, the data appear to support the coexistence of vastly different dynamics within the condensates. These observations were first made and rationalized in a study that dissected the dynamics of model proteins and RNA molecules in synthetic condensates. Please see: <https://www.pnas.org/content/early/2019/03/28/1821038116>. This work is also relevant in the context of the data presented in Figure 2. A clear interpretation of the observations comes when one considers the effects of dynamically arrested phase separation. This would explain why chromatin shows territorial organization in some cases but not others. As written, the message gleaned might suggest that this is an equilibrium phenomenon, but it is most likely the result of slow dynamics. The work of Boeynaems et al. provides some context.
6. In the filter binding assays, the abscissa show protein concentration as the parameter. However, this should be activity of the protein because SAM undergoes polymerization, and the extent of polymerization depends very much on the construct. Therefore, there are likely to be

polymeric linkage effects that have to be accounted for when analyzing the data. Does DNA bind to SAM polymers? If yes, do we know the relative affinities of DNA to monomeric SAMs vs. polymers of SAM? Alternatively, can one assess the concentration of free monomers? If yes, the analysis should be performed in terms of concentrations of free SAM monomers and this would provide a clearer assessment of the binding data. If such analyses are difficult to perform, then this should be noted up front because the apparent K_d values are difficult to interpret without knowledge of the activity of SAM. These issues are especially relevant given that the apparent K_d values do not (rather mysteriously) affect the driving forces for phase separation, although the extent of SAM polymerization presumably does affect DNA binding. There is a mystery lurking here and at this juncture there is a lack of clarity regarding the linkage among binding, polymerization, and the driving forces of phase separation driven by a combination of homotypic and heterotypic interactions. Comments are also made about smaller sizes and "fragility" of condensates formed by mutations that weaken / disrupt SAM polymerization. However, it is not clear how "fragility" is quantified or what it means in this context. Please clarify.

7. The ability of a construct to undergo phase separation in the presence of a crowder is predicated on the extent of exclusion of the crowder from the dense phase. It is very difficult to obtain an assessment of the intrinsic driving forces mediated by effectively homotypic interactions from measurements in the presence of crowders. In my biased view, columns two and the in Figure 3E are likely to be misleading and open to an assortment of interpretations because the effects of crowders are not fully understood and rather complex. My suggestion would be to delete these two columns and to expunge mention of these in the main text. My biased view is that these data do not add value and are refractory to a clear interpretation.

8. The narrative asserts that the concentrations of nucleosomes in the three different condensates are the same according to Figure 3G. The data seem to suggest otherwise. Please clarify. Likewise, the narrative asserts that the sizes of condensates formed by ML and EH are smaller than those formed by WT. However, the data in Figure 3H suggest that it is the EH construct that forms discernibly smaller condensates. It is worth noting that it is difficult to interpret the implications of changes to condensate size when data are presented for a single set of conditions and the images are collected at single time points. The takeaway for the readers is unclear here. What physical interpretations do the authors draw from the data in Figures 3G-J? Please clarify. A similar comment applies to the observation regarding changes in sign to the condensates upon their incubation with nuclear extracts. Why do the condensates change in size? This is an important question but it cannot be answered without considering linkage effects - a rudimentary version of this is available in a recent publication - <http://www.jbc.org/content/293/10/3734>.

9. The data regarding changes to lysine accessibilities are interesting. However, their inclusion is a bit of mystery because a clear takeaway from these data does not come through. There are site specific changes for sure. Should these be used to construct a molecular level understanding of the organization within condensates? This is not clear. Please clarify. The assay itself is very interesting and exciting. And it would help immensely if the interpretations can be made quantitative.

10. One appreciates the data showing that Mini-Ph condensates can recruit components of the Prc complex and that there is a discernible enhancement of ubiquitylation activity. In the current narrative these data are presented without a quantitative analysis of why these observations should result from condensate formation. There are numerous aspects that are convolved into observations of selective partitioning. Further, it does not have to follow that condensates will necessarily increase protein / enzymatic activity. Such observations are often predicated on the way the data are obtained and how they are analyzed. Perhaps the insertion of suitable caveats and / or quantitative modeling of the data and why these inferences may or may not be valid would help. In this context, I appear to have missed the evidence for enhanced ubiquitylation in vivo although the correlation is mentioned in the text. Please clarify.

11. The interplay between Ph polymerization and condensation are not well developed. The discussion appears to point away from the findings in the current study. There are two possibilities: SAM polymerization and Ph condensation are either orthogonal or they are synergistic. There is a clear convergence upon the idea of multivalence of motifs / stickers as the direct contributors to phase separation. Multivalence that drives phase separation can come about in two different ways - either this multivalence is intrinsic to the system or the multivalence is an emergent consequence of multimerization / polymerization of domains that are distinct from the stickers that drive condensate formation. Please see <https://www.annualreviews.org/doi/abs/10.1146/annurev-biophys-121219-081629>. Of course, a synergy between intrinsic and emergent multivalence is also likely. To put the findings in the context of what is currently known, it would help to call out the concepts of multivalence directly and explain how the findings might fit in this context.

12. The discussion offers tantalizing suggestions regarding the linkers. In this context (see point 1) it would help if the effective solvation volumes of the linker orthologs were analyzed or at least the linker sequences across orthologs that support the statements of hypervariability were furnished in the SI.

13. This is an earnest misunderstanding / question: Is O-linked glycosylation relevant in the nucleus? One often thinks of such effects as being relevant to secreted proteins or proteins in the extracellular milieu.

14. With all due respect, the PPPS mechanism is purely speculative and at this juncture seems rather like a schematic absent any quantitative support. Even in the current narrative, a lot of the pronouncements around the proposal of PPPS are very qualitative. It is not clear that the inclusion of these statements adds value to the discussion section. Respectfully, I would propose that the inclusion of these schematic ideas in the discussion elevates unproven ideas and could engender the misconception that the data somehow provide "proof" of this somewhat unclear concept of PPPS.

To conclude this rather long review, I submit that this work is important, interesting and highly relevant. It will be seen as a valuable addition to the phase separation literature. Prior to publication it is in need of clarifications, revisions, and some rethinking.

We are grateful to both reviewers for their thoughtful comments. We have substantially revised the text of the manuscript, revised several of the main figures (Fig. 1-4), and added 8 new Supplementary figures. We believe these changes have improved the manuscript, and have detailed them in addressing each comment below. Please note that we have appended the manuscript with tracked changes and line numbering to facilitate navigating the changes (line numbers apply to when track changes is hidden).

REVIEWER COMMENTS

Reviewer #1 (Remarks to the Author):

This paper well-written paper presents a thoughtful analysis of the ability of a key part of the PH protein to generate phase separated structures. PH is a key component of the Polycomb-Group. It is part of the PRC1 complex and has been previously shown to be involved in long range interactions involving its SAM domain. There is, therefore, strong justification for studying its role in phase separation given the potential connection between higher order networking and phase separation.

Strengths of the paper are that the experiments are carefully done and not over-interpreted. Multiple models are thoughtfully considered in the discussion. There is compelling data presented showing that the mini-PH protein can phase separate in vitro to concentrate template, and interesting analysis of mixing of templates (Fig. 2) and a nice demonstration that other PRC1 components can partition into the phase separated droplets. Ubiquitylation studies are used to validate that the PRC1 in droplets is functional, and cell culture studies show that droplets can form in cells. Overall, this is an interesting paper that contributes to the field and is useful in further understanding polycomb phase separation and enzymatic activity and the complexities of multivalent, intramolecular and intermolecular interactions that have a significant role in epigenetics. I saw no significant weaknesses and offer a few comments for consideration, mainly on presentation.

Comments:

1) In terms of organisation, there were a few issues with the model that were thoroughly addressed in the discussion but lacked nuance in the results section. Introduction of some of these complexities earlier would be helpful to the reader for clarity purposes (for example, the conflicting views of ncPRC1 and cPRC1 ubiquitylation activity).

Response: We have expanded the description of nc and cPRC1 in the introduction (lines 25-38), and attempted to state previous results that are directly relevant to the data presented here more clearly.

2) Figures could be more clearly referenced – there were several times the text brought up questions that were then answered separately by a closer look at the figures but this could be mitigated by increased clarity in figure references throughout the results section.

Response: We have increased the figure citations.

3) If complex coacervation is to be discussed, analysis of protein charge should be included in the first section of the results.

Response: We have removed the statement about complex coacervation, concurring with both reviewers that we have not provided evidence to support it.

4) Lysine accessibility is analyzed with respect to mini-PH, but it seems unclear how the rest of the protein would affect this data? In addition, it would be helpful to see mini-PH's behavior in vivo as compared to the full protein? This experiment needs some qualification in the text as to what conclusions can be reached and what caveats there are when thinking about the natural context of these domains.

Response: We have not carried out the lysine accessibility assays with the full length protein, or in the context of PRC1 (which could be done with Mini-Ph since it can assemble into PRC1). We are very interested in these experiments but feel they are outside the scope of the current analysis. We have expanded this assay to a polymerization defective Mini-Ph (in revised Fig. 4, Supplementary Fig. 13), which we believe clarifies the interpretation of the data, at least with respect to Mini-Ph. We have added a statement (lines 277-279) to point out the caveat that the other sequences in Ph or assembly into PRC1 could change the organization of the Mini-Ph region.

The question about the activity of Mini-Ph in vivo is a good one; we struggled with including or not including these data in the initial submission. We have included live imaging experiments of Venus-Mini-Ph and Venus-Mini-Ph Δ SAM (new Supplementary Figure 18, lines 395-400). In cells, Mini-Ph does not form the multiple foci observed when Ph is overexpressed (as in Figure 7). It either does not form foci, or forms a single focus, which can be very large. We have done this experiment many times, using promoters that drive different levels of expression, and we do not observe cells with multiple Mini-Ph foci. Because full length Ph forms multiple foci, but only when the SAM is present, we conclude that the other sequences in Ph work with the SAM in vivo. In addition to the Mini-Ph region, Ph contains 3 distinct IDRs; we are in the process of analyzing the role of each of them. From our initial analysis, it is clear that each of these IDRs affects the behavior of Ph in cells, but only when the SAM is present. While we believe that these experiments are essential to understanding Ph function vis-a-vis phase separation and other mechanisms, substantially more work is needed. By including the in vivo data on Mini-Ph, we are able to acknowledge this complexity, and avoid misleading the reader into thinking that Mini-Ph is the whole story.

5) It would be interesting to understand the conclusions from 6B and 6C at multiple concentrations of protein (WT or PH mutant input). In addition, how is protein concentration of the recruited extract being normalized between the samples? For example, is ubiquitylation activity in the polymerization mutant equivalent to less concentrated WT sample? This might suggest polymerization ability is linked to recruiting ability? It is not clear what is meant by fragile in this context – are you suggesting that the same amount is recruited but then activity is still lower because interactions are less robust?

Response: We have not used multiple concentrations to form condensates in the experiments in Figure 6B/C because the experiment is already quite complicated. Because condensate formation is sensitive to the ratio of Mini-Ph to chromatin, it is not so clear to us how to interpret a protein titration. To normalize the extract among samples, each experiment is set up so that all three mutants form condensates with the same chromatin template, and then the same amount of the same nuclear extract preparation is used so that the conditions should be identical for each set of condensates. It is possible

that condensates formed by the mutants recruit less PRC1. As pointed out, we did not test this. We have added a statement clarifying this. We have removed the adjective “fragile” in our description of Mini-Ph EH-chromatin condensates because both reviewers found it to be imprecise. We have also substantially expanded our characterization of the polymerization mutants, which indicates that condensates formed by them are more sensitive to NaCl and ATP (Supplementary Fig. 8 and 9). We also now show that Mini-Ph EH is more mobile in condensates (Supplementary Fig. 14) We have attempted to clarify the text describing the mutants and their implications for the mechanisms underlying phase separation (summarized in the model figure we now include as Supplementary Fig. 10, also lines 220-235).

Reviewer #2 (Remarks to the Author):

First and foremost, I offer my sincere and unvarnished apologies to the authors for my excessive tardiness in turning this review around. The situation brought on by Covid-19 and the massive ramp up in responsibilities, especially teaching and mentoring, have taken a significant toll on bandwidth. While this is a legitimate excuse, I recognize that it will not and should not assuage the authors sense of grievance.

In this work, the authors focus on the phase behavior of a truncated version of Polyhomeotic (Mini-Ph) with chromatin. The hypothesis is that the transcriptional repression is organized by the formation of protein-DNA condensates. To test this hypothesis, the authors assess the condensate forming ability of Mini-Ph with chromatin. They query the internal dynamics of molecules within these condensates, the exchange of material across condensates, and the fusion dynamics of condensates. Based on deletion mutagenesis experiments the authors show that the sterile alpha motif, a polymerization domain, is essential for forming condensates, although this does not appear to derive from the polymerization activity of SAM - which is surprising. Further characterizations of lysine acetylation and histone ubiquitylation are provided.

Overall, the experiments are interesting, well designed, and well thought out. The conclusions stay confined to the constraints imposed by the data and the manuscript certainly merits publication following some key revisions listed below.

1. Please provide a summary of the sequence features of the unstructured linker.

Response: We have added Supplementary Figure 4 and Supplementary Table 1 (lines 100-120), which provide the annotated sequence of the Mini-Ph linker, a basic analysis of the sequence properties of the *Drosophila* and human linkers, and sequence alignments showing that the two *Drosophila* orthologues of Ph are well conserved, but the three human orthologues are very different from the *Drosophila* linkers. We have further highlighted previous data analyzing the effect of different linkers on limiting or promoting Mini-Ph polymerization (Robinson et al., doi: 10.1074/jbc.M111.336115; Robinson et al., doi: 10.1021/bi3004318.XX). Of course, it would be very interesting to test conditions under which the human PHCs undergo phase separation, which we have not yet done.

2. Please note that Figure 1G is not a phase diagram. I respect the qualifier of this being a qualitative

phase diagram, but nevertheless request that this be referred to as a limited coarse-grained delineation of the boundary between the one- and two-phase regimes.

Response: We have removed the term “phase diagram” as requested, slightly expanded the explanation of the experiment in the text, and incorporated the recommended language for describing these results.

3. To invoke complex coacervation as the mechanism of phase separation, one would have to establish that the central and perhaps only contribution to the driving forces for phase separation are the extent of mismatches between charges on the DNA and protein. This would require a systematic assessment of the effects of charge imbalance by titrating salt and pH in different combinations and investigating how the critical point is crossed as a function of these titrations. That would be too much to ask. The simpler fix is to indicate that the mechanism likely involves a combination of homotypic and heterotypic interactions. If and only if the mechanism is solely driven by complementary electrostatic interactions and the release of condensed counterions can one invoke a purely complex coacervation mechanism.

Response: We have removed the statement about complex coacervation, in response to both reviewers. We have also expanded our discussion of the interactions driving phase separation (see response to points 6 and 11 below).

4. The estimate of intra-condensate concentrations of nucleosomes will be predicated on the standard curve used to calibrate these measurements. How reliable are such calibrations for measuring concentrations in dense phases? The 60x mismatch between calibrations based on free dye vs. the approach used here would appear to suggest that there might be photophysical artifacts. What is unclear is if the current approach does not rely on the use of labeled molecules. Please clarify.

Response: We are using the labelled histone octamers to prepare calibration curves. These are the same histone octamers that are deposited on DNA to prepare the chromatin templates. For this reason we think they are more comparable than free Cy3 dye (it should be noted that by free Cy3, we mean unreacted NHS-Cy3, the reagent used to label the histone proteins). We believe that photophysical artifacts are responsible for the underestimation of free Cy3 as measured by microscopy. We have quantified the Cy3 concentration (i.e. labelling efficiency) on our histone proteins both by nanodrop and by SDS-PAGE (and a Typhoon imager to quantify), using free dye as the standard. These measurements are consistent between the two methods. However, if we titrate free Cy3 next to Cy3 labelled histone octamers and measure the concentration based on intensity from images, the intensity measured from images of free Cy3 is much lower (at least 60X) than the intensity measured from the same concentration of labelled histone octamers. We think the histone octamer titration is relevant for measuring the Mini-Ph-chromatin dense phase because the intensities of the dense phase are only about 3X higher than the top point in our titration. We also adapted the strategy of Gibson et al. (2019, Cell) of measuring mixtures of labelled and unlabelled chromatin. We tested a 1:10 mix of labelled to unlabelled and found that the corrected value for the 1:10 mixture was similar (although slightly higher) to the value obtained with only labelled chromatin (28 versus 21 μ M). We now include this experiment in Supplementary Fig. 5E.

5. Figure 2B should be reanalyzed to quantify the contributions of mobile vs. immobile fractions at

different time points. This is necessary since the recovery is not 100%. Further, the traces for chromatin are not shown here. This is important to include in order to support the observations made in the text. Also, the data appear to support the coexistence of vastly different dynamics within the condensates. These observations were first made and rationalized in a study that dissected the dynamics of model proteins and RNA molecules in synthetic condensates. Please see: <https://www.pnas.org/content/early/2019/03/28/1821038116>. This work is also relevant in the context of the data presented in Figure 2. A clear interpretation of the observations comes when one considers the effects of dynamically arrested phase separation. This would explain why chromatin shows territorial organization in some cases but not others. As written, the message gleaned might suggest that this is an equilibrium phenomenon, but it is most likely the result of slow dynamics. The work of Boeynaems et al. provides some context.

Response: We agree that Mini-Ph and chromatin (or DNA) show very different dynamics in condensates. in FRAP experiments. We did not understand how the immobile fraction should be quantified, and so have not been able to do this. We have included a trace from the chromatin, which we agree is important to show the distinct dynamics. We have revised our description of these experiments to highlight the difference in dynamics. We are not sure if this reflects similar processes as described in Boeynaems et al. for small RNA molecules (where RNA-RNA interactions lead to slow dynamics of the RNAs). However, we introduce the possibility that nucleosome-nucleosome interactions (which are likely to occur in our system), could be contributing to the slow or arrested dynamics of chromatin. We think it is also possible that the large size of our chromatin templates (~55 nucleosomes), and the large number of Mini-Phs bound to each template is playing a role, and that these large chromatin templates may behave more like regions of chromosomes, which show slow or no recovery in FRAP experiments. We have added these considerations to the text (lines 150-160).

6. In the filter binding assays, the abscissa show protein concentration as the parameter. However, this should be activity of the protein because SAM undergoes polymerization, and the extent of polymerization depends very much on the construct. Therefore, there are likely to be polysteric linkage effects that have to be accounted for when analyzing the data. Does DNA bind to SAM polymers? If yes, do we know the relative affinities of DNA to monomeric SAMs vs. polymers of SAM? Alternatively, can one assess the concentration of free monomers? If yes, the analysis should be performed in terms of concentrations of free SAM monomers and this would provide a clearer assessment of the binding data. If such analyses are difficult to perform, then this should be noted up front because the apparent K_d values are difficult to interpret without knowledge of the activity of SAM. These issues are especially relevant given that the apparent K_d values do not (rather mysteriously) affect the driving forces for phase separation, although the extent of SAM polymerization presumably does affect DNA binding. There is a mystery lurking here and at this juncture there is a lack of clarity regarding the linkage among binding, polymerization, and the driving forces of phase separation driven by a combination of homotypic and heterotypic interactions. Comments are also made about smaller sizes and "fragility" of condensates formed by mutations that weaken / disrupt SAM polymerization. However, it is not clear how "fragility" is quantified or what it means in this context. Please clarify.

11. The interplay between Ph polymerization and condensation are not well developed. The discussion appears to point away from the findings in the current study. There are two possibilities: SAM polymerization and Ph condensation are either orthogonal or they are synergistic. There is a clear convergence upon the idea of multivalence of motifs / stickers as the direct contributors to phase separation. Multivalence that drives phase separation can come about in two different ways - either this multivalence is intrinsic to the system or the multivalence is an emergent consequence of multimerization / polymerization of domains that are distinct from the stickers that drive condensate formation. Please see <https://www.annualreviews.org/doi/abs/10.1146/annurev-biophys-121219-081629>. Of course, a synergy between intrinsic and emergent multivalence is also likely. To put the findings in the context of what is currently known, it would help to call out the concepts of multivalence directly and explain how the findings might fit in this context.

Response: We have combined the response to these two comments, which we think address a related issue. We appreciate these comments very much, particularly the underlined points, which we think identify key ambiguities in our initial submission. This has prompted us to re-think the results and (we hope) explain the data more carefully. There are several points to be made, which we have done by modifying text and the figures, and adding additional figures (Supplementary Fig. 8, 9, 10, 14)

First, we have clarified two points about the SAM and the expected oligomeric state of wild type and mutant Mini-Ph in the text. At the concentrations of protein used for phase separation (5 μ M), the SAM will be self-associated. However, the polymer-limiting effect of the Mini-Ph linker must also be considered. As shown previously using AUC (see Fig. 3 of Robinson et al., JBC 2012), Mini-Ph exists mainly as short polymers (4-6 units), and does not polymerize further even at 10x higher concentrations. Thus, dynamic polymerization mediated by Ph SAM is unlikely to contribute to phase separation under our in vitro conditions. Instead, SAM polymerization is likely increasing the driving forces for phase separation by 1) creating multivalency by clustering the FCS domains; 2) increasing the affinity for DNA/chromatin. We have explained this argument, and added a schematic to Figure 1 (Figure 1B) to emphasize that Mini-Ph exists as short polymers prior to binding chromatin and cannot further polymerize.

We have also added additional data (Supplementary Fig. 8 and 9) regarding the EH and ML mutants, which show that both are more sensitive to NaCl, and the EH mutant is more sensitive to ATP, consistent with lower driving forces for phase separation in the mutants (lines 210-235). We have expanded our analysis of the EH mutant, showing that it is more mobile in condensates (by FRAP, Supplementary Fig. 14) (lines 288-304), consistent with the lower DNA binding affinity, and weaker homotypic interactions. We have also carried out the acetylation footprinting assay with the EH mutant (detailed in response to Point 4 of rev. 1) (Fig. 4D, E, Supplementary Fig. 13). This shows that the accessibility of the SAM, and indeed of the whole of Mini-Ph is different in wild type (polymerized) and EH mutant (monomeric) (lines 265-286). Finally, we have explicitly considered what the interactions underlying phase separation in our system are likely to be by providing a schematic model (Supplementary Figure 10). This is framed as attempting to explain why Mini-Ph EH phase separates, while Mini-Ph Δ SAM does not (i.e. why the SAM but not its polymerization activity is required), even though both bind DNA with similar affinities (Fig. 3). We highlight a previously identified interaction between the linker and the SAM (present in Mini-Ph-EH, but not in Mini-Ph Δ SAM), and possible weak SAM-SAM interactions.

Unfortunately we were not able to explicitly address the question about the DNA binding assays. At the low concentrations used for DNA binding (well below the measured SAM-SAM K_d of 200nM), we expect Mini-Ph might dissociate into monomers. However, this does not easily explain the much lower K_d of Mini-Ph than either Mini-Ph-EH or Mini-Ph Δ SAM. We think it is likely that the K_d for SAM-mediated Mini-Ph-Mini-Ph interactions may be lower than measured for the isolated SAM, so that the DNA binding we measure is for the oligomer.

7. The ability of a construct to undergo phase separation in the presence of a crowder is predicated on the extent of exclusion of the crowder from the dense phase. It is very difficult to obtain an assessment of the intrinsic driving forces mediated by effectively homotypic interactions from measurements in the presence of crowders. In my biased view, columns two and the in Figure 3E are likely to be misleading and open to an assortment of interpretations because the effects of crowders are not fully understood and rather complex. My suggestion would be to delete these two columns and to expunge mention of these in the main text. My biased view is that these data do not add value and are refractory to a clear interpretation.

Response: We have removed these data as requested.

8. The narrative asserts that the concentrations of nucleosomes in the three different condensates are the same according to Figure 3G. The data seem to suggest otherwise. Please clarify. Likewise, the narrative asserts that the sizes of condensates formed by ML and EH are smaller than those formed by WT. However, the data in Figure 3H suggest that it is the EH construct that forms discernibly smaller condensates. It is worth noting that it is difficult to interpret the implications of changes to condensate size when data are presented for a single set of conditions and the images are collected at single time points. The takeaway for the readers is unclear here. What physical interpretations do the authors draw from the data in Figures 3G-J? Please clarify. A similar comment applies to the observation regarding changes in sign to the condensates upon their incubation with nuclear extracts. Why do the condensates change in size? This is an important question but it cannot be answered without considering linkage effects - a rudimentary version of this is available in a recent publication - <http://www.jbc.org/content/293/10/3734>.

Response: We have changed the text regarding Figure 3G to state that the nucleosome concentration is “similar”, in accordance with the p-values, which do not point to a clear difference. We do not know why the condensates change in size, and we have added a statement to the text to indicate this. We expect that it is due to other proteins or nucleic acids in the extracts, which might compete for binding, or otherwise change the properties of condensates. Indeed, our preliminary data indicates that PRC1 can change the size and properties of Mini-Ph-chromatin condensates. We have added consideration of why the condensates are smaller to the text (lines 320-326), but have not invoked a specific mechanism (for example changes in oligomeric state) because we do not have evidence.

9. The data regarding changes to lysine accessibilities are interesting. However, their inclusion is a bit of mystery because a clear takeaway from these data does not come through. There are site specific changes for sure. Should these be used to construct a molecular level understanding of the organization

within condensates? This is not clear. Please clarify. The assay itself is very interesting and exciting. And it would help immensely if the interpretations can be made quantitative.

Response: We agree (also with reviewer 1, see point 4) that the lysine accessibility data in the first submission have a clear interpretation with respect to DNA binding (i.e. protection of residues in the FCS domain), but not with respect to phase separation/SAM polymerization. We have repeated this assay with the EH mutant, which should not polymerize (Fig. 4D, E; Supplementary Fig.13). We think these data provide insight into the role of SAM polymerization in organizing Mini-Ph. We have also included additional quantification of accessibility (Fig. 4E, Supplementary Fig. 11C, 13D-F). Finally, the publication describing the assay is now cited.

10. One appreciates the data showing that Mini-Ph condensates can recruit components of the Prc complex and that there is a discernible enhancement of ubiquitylation activity. In the current narrative these data are presented without a quantitative analysis of why these observations should result from condensate formation. There are numerous aspect that are convolved into observations of selective partitioning. Further, it does not have to follow that condensates will necessarily increase protein / enzymatic activity. Such observations are often predicated on the way the data are obtained and how they are analyzed. Perhaps the insertion of suitable caveats and / or quantitative modeling of the data and why these inferences may or may not be valid would help. In this context, I appear to have missed the evidence for enhanced ubiquitylation in vivo although the correlation is mentioned in the text. Please clarify.

Response: We agree that our interpretation of the increase in ubiquitylation activity was ambiguous, because we do not know why the activity is increased. Combining these data with the observations from the pelleting assay in nuclear extracts (showing that PRC1 is recruited to condensates), suggests that the change in enzyme activity could simply reflect an effective increase in concentration of PRC1, or other components of the ubiquitylation machinery (we we did not test). However, at least for the reconstituted in vitro experiments (Fig. x) we think this is not likely to be the explanation for several reasons: 1) PRC1 binds DNA and chromatin very tightly (subnanomolar affinity for 150bp DNA), so that even at the lowest concentrations used, binding affinity is unlikely to limit the E3 ligase activity. 2) the other components required for ubiquitylation were intentionally used at saturating concentrations. 3) we observe a ~2x increase in enzyme activity, yet chromatin is concentrated at least 10x in condensates (from our measured chromatin concentration, in which nucleosomes are at 150 nM in solution and 21 μ M in condensates). Indeed, we think it is likely that the reaction rate, or processivity are enhanced in condensates. This is consistent with a recent publication indicating that chromatin condensation facilitates histone ubiquitylation in vivo, as well as data implicating phase separation of the machinery for ubiquitylation of H2B in enhancing ubiquitylation, which we now cite. However, at this time, we do not have any evidence for which step(s) in the reaction are affected. We have clarified the discussion to address these ambiguities, as well as pointing out that enhancement of ubiquitylation might not have been the anticipated result (lines 508-522).

12. The discussion offers tantalizing suggestions regarding the linkers. In this context (see point 1) it

would help if the effective solvation volumes of the linker orthologs were analyzed or at least the linker sequences across orthologs that support the statements of hypervariability were furnished in the SI.

Response: Please see response to point 1.

13. This is an earnest misunderstanding / question: Is O-linked glycosylation relevant in the nucleus? One often thinks of such effects as being relevant to secreted proteins or proteins in the extracellular milieu.

Response: There is quite an extensive literature on O-linked glycosylation of nuclear proteins (including nuclear pore proteins). It is a field that is not without controversy, but genetic data linking ogt, the gene encoding the enzyme that transfers O-GlcNAc to nuclear proteins, including Ph, are solid. These two reviews give a broader perspective on the function of the modification in the nucleus, and the other two citations explicitly link the modification to Ph function (via the SAM in the case of the second publication).

Lewis BA, Hanover JA. O-GlcNAc and the epigenetic regulation of gene expression. *J Biol Chem*. 2014;289(50):34440-34448. doi:10.1074/jbc.R114.595439

Hanover, J., Krause, M. & Love, D. linking metabolism to epigenetics through O-GlcNAcylation. *Nat Rev Mol Cell Biol* **13**, 312–321 (2012). <https://doi-org.proxy3.library.mcgill.ca/10.1038/nrm3334>.

Gambetta MC, Müller J. O-GlcNAcylation prevents aggregation of the Polycomb group repressor polyhomeotic. *Dev Cell*. 2014;31(5):629-639. doi:10.1016/j.devcel.2014.10.020

Gambetta MC, Oktaba K, Müller J. Essential role of the glycosyltransferase *sxc/Ogt* in polycomb repression. *Science*. 2009;325(5936):93-96. doi:10.1126/science.1169727

14. With all due respect, the PPPS mechanism is purely speculative and at this juncture seems rather like a schematic absent any quantitative support. Even in the current narrative, a lot of the pronouncements around the proposal of PPPS are very qualitative. It is not clear that the inclusion of these statements adds value to the discussion section. Respectfully, I would propose that the inclusion of these schematic ideas in the discussion elevates unproven ideas and could engender the misconception that the data somehow provide "proof" of this somewhat unclear concept of PPPS.

Response: We are a bit stymied by this comment since the discussion of PPPS as a mechanism that is relevant for chromatin is prevalent and (in our view) carefully explained by theoreticians and experimentalists (e.g. Erdel et al., *Mol. Cell*, 2020, Erdel & Rippe, 2019, Hildebrand & Dekker, 2020) (lines 482-486). The idea of nucleosome bridging leading to collapse of regions of the chromatin polymer was also used to explain simulations of the effect of the ML mutation in Ph in vivo (Wani et al., *Nat. Comm.* 2016). We have removed the term "PPPS", and made certain to cite relevant explanations of this model.

To conclude this rather long review, I submit that this work is important, interesting and highly relevant.

It will be seen as a valuable addition to the phase separation literature. Prior to publication it is in need of clarifications, revisions, and some rethinking.

REVIEWERS' COMMENTS

Reviewer #1 (Remarks to the Author):

This revision did a nice job of responding to my comments. The revised paper has improved detail on polycomb mechanisms and the activity of cPRC1 and ncPRC1. The introduction is more clear in describing the variety of polycomb complexes and their known and speculated roles in chromatin organization. The increased figure citations improve the clarity of the manuscript throughout the results section. Removal of the claim of complex coacervation improves the overall evidence based characterization of LLPS and the strength of the paper's conclusions.

The data and explanation for mini-Ph behavior has been improved. Addition of the polymerization defective mini-Ph EH allows for increased clarity in drawing conclusions from the lysine accessibility experiments. It is reasonable that mini-Ph cannot be directly compared with the full-length Ph in vitro - the full length Ph is difficult to purify. The inclusion of the mini-Ph in vivo data allows for increased transparency regarding the complex relationship between in vitro phase separation characterization and in vivo activity. This presents a clear future direction for their work but is not necessary for the completeness of the current story.

The addition of the statement regarding PRC1 recruitment clarifies the presentation of this data. The removal of "fragile" as a descriptor is satisfactory and the addition of the sensitivity to ATP and NaCl experiments further strengthens the characterization of condensates. The increased description of the Ph mutants and their hypothesized effects on phase separation and polymerization activity is useful to the reader for interpretation of the data. Buffering capacity of phase separation may explain the results in Fig 5B and 5C showing differences in condensate size between buffer and nuclear extracts (<https://science.sciencemag.org/content/367/6476/464.abstract>). Additionally, the increased number of PRC1 gene targets present in nuclear extracts may drive the increasing number of condensates as a speculative mechanism in line with the authors' hypothesis that nucleic acids may be responsible for this change.

Reviewer #2 made interesting points in criticizing the manuscript from a biophysical perspective and the response to that review improved the quality of the manuscript from my standpoint as well.

I strongly support publication as is.

Reviewer #2 (Remarks to the Author):

The authors have addressed all the comments I raised and done so satisfactorily. I have no further revisions to request.

REVIEWERS' COMMENTS

Reviewer #1 (Remarks to the Author):

This revision did a nice job of responding to my comments. The revised paper has improved detail on polycomb mechanisms and the activity of cPRC1 and ncPRC1. The introduction is more clear in describing the variety of polycomb complexes and their known and speculated roles in chromatin organization. The increased figure citations improve the clarity of the manuscript throughout the results section. Removal of the claim of complex coacervation improves the overall evidence based characterization of LLPS and the strength of the paper's conclusions.

The data and explanation for mini-Ph behavior has been improved. Addition of the polymerization defective mini-Ph EH allows for increased clarity in drawing conclusions from the lysine accessibility experiments. It is reasonable that mini-Ph cannot be directly compared with the full-length Ph in vitro - the full length Ph is difficult to purify. The inclusion of the mini-Ph in vivo data allows for increased transparency regarding the complex relationship between in vitro phase separation characterization and in vivo activity. This presents a clear future direction for their work but is not necessary for the completeness of the current story.

The addition of the statement regarding PRC1 recruitment clarifies the presentation of this data. The removal of "fragile" as a descriptor is satisfactory and the addition of the sensitivity to ATP and NaCl experiments further strengthens the characterization of condensates. The increased description of the Ph mutants and their hypothesized effects on phase separation and polymerization activity is useful to the reader for interpretation of the data. Buffering capacity of phase separation may explain the results in Fig 5B and 5C showing differences in condensate size between buffer and nuclear extracts (<https://science.sciencemag.org/content/367/6476/464.abstract>). Additionally, the increased number of PRC1 gene targets present in nuclear extracts may drive the increasing number of condensates as a speculative mechanism in line with the authors' hypothesis that nucleic acids may be responsible for this change.

Reviewer #2 made interesting points in criticizing the manuscript from a biophysical perspective and the response to that review improved the quality of the manuscript from my standpoint as well.

I strongly support publication as is.

Reviewer #2 (Remarks to the Author):

The authors have addressed all the comments I raised and done so satisfactorily. I have no further revisions to request.